# Resolving Discrepancies in Compute-Optimal Scaling of Language Models

Tomer Porian[*]   Mitchell Wortsman[†]   Jenia Jitsev[‡]   Ludwig Schmidt[†]   Yair Carmon[*]

## Abstract

Kaplan et al. [30] and Hoffmann et al. [25] developed influential scaling laws for the optimal model size as a function of the compute budget, but these laws yield substantially different predictions. We explain the discrepancy by reproducing the Kaplan et al. scaling law on two datasets (OpenWebText2 and RefinedWeb) and identifying three factors causing the difference: last layer computational cost, warmup duration, and scale-dependent optimizer tuning. With these factors corrected, we obtain excellent agreement with the Hoffmann et al. (i.e., "Chinchilla") scaling law. Counter to a hypothesis implied in Hoffmann et al. [25], we find that careful learning rate decay is not essential for the validity of their scaling law. As a secondary result, we derive scaling laws for the optimal learning rate and batch size, finding that tuning the AdamW $\beta_2$ parameter is essential at lower batch sizes.

## 1  Introduction

We consider the problem of compute-optimal language model training: given a compute budget $C$, we wish to predict how to best allocate it across model size (in parameters) and dataset size (in tokens). With pretraining budgets ever-increasing, compute-optimal scaling is a question of paramount importance. In their seminal work, Kaplan et al. [30] proposed a scaling law predicting that the optimal ratio of tokens to parameters decays as a power of $C$. This scaling law was influential in determining the size of GPT-3 and several subsequent models [12, 51, 43, 32, 47, 62, 52]. However, Hoffmann et al. [25] challenged its validity, arguing instead that the optimal token-to-parameter ratio should be approximately independent of $C$, and that contemporary models had too many parameters relative to their number of training tokens. Based on this prediction, they trained a 67B parameters model called Chinchilla and which outperformed larger models with a similar compute budget.

While Hoffmann et al. [25] and subsequent work [55, 56, 17, 26, 15] established that following the Hoffmann et al. scaling law leads to better performance than Kaplan et al. scaling, it is still important to understand *why* the two works arrived at different conclusions. Is the difference due to architecture, training setup, pretraining data, results analysis, or perhaps something else entirely? The answer could teach us important lessons on how to correctly predict and perform model scaling.

Hoffmann et al. [25] hypothesize that the scaling law discrepancy is due to Kaplan et al. [30] not tailoring the learning rate schedule for each token budget separately. While they demonstrate that mismatched learning rate decay results in a higher loss, they do not show it leads to a different compute-optimal scaling law. We further discuss Hoffmann et al. [25]'s hypothesis in Appendix A. To the best of our knowledge, this hypothesis is the only explanation offered in the literature so far.

**Our contribution.**   In this work, we uncover three factors contributing to the discrepancy, and disprove Hoffman et al.'s hypothesis about the role of learning rate decay; Figure 1 illustrates our main

---

[*]Tel Aviv University; correspondence to `tomerpor@gmail.com` and `ycarmon@tauex.tau.ac.il`.

[†]University of Washington.

[‡]Jülich Supercomputing Centre (JSC) and LAION.

38th Conference on Neural Information Processing Systems (NeurIPS 2024).

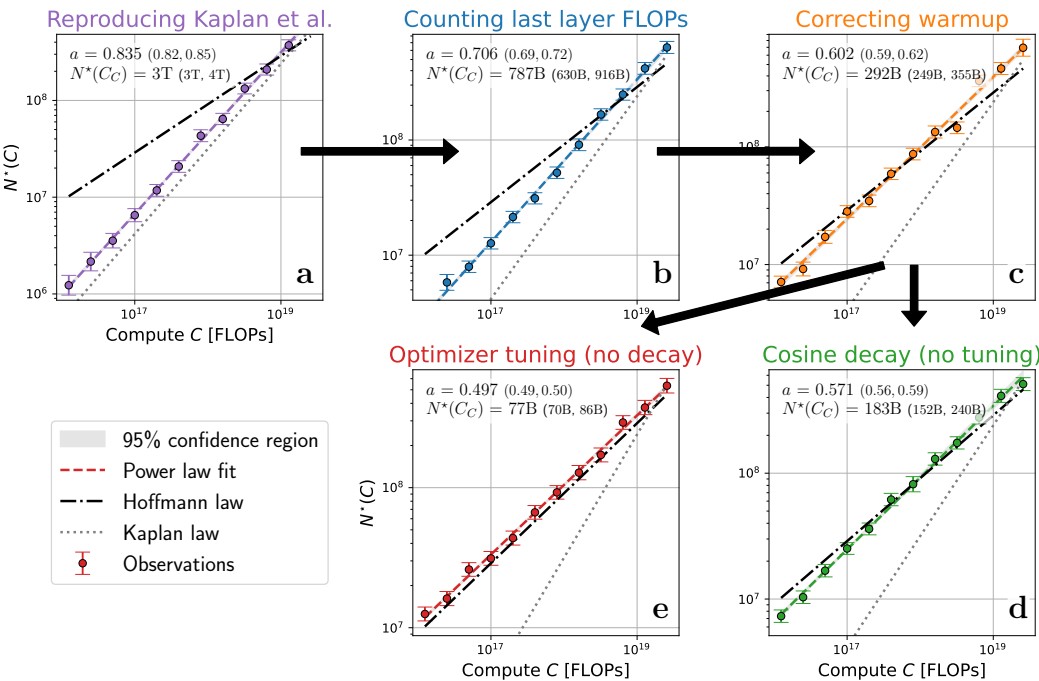

Figure 1: By analyzing over 900 training runs, we uncover the factors leading to the discrepency between the scaling laws of Kaplan et al. (panel a) and Hoffmann et al. (panel e). Each panel shows observations of the optimal model size $N^\star$ as a function of the compute budget $C$, as well as power law fits of the form $N^\star(C) \propto C^a$. Labels show point estimates and 95% confidence intervals for $a$ and for the optimal model at $C_C = 5.88\mathrm{e}23$, the compute budget used for training Chinchilla.

results. We begin by reproducing the Kaplan et al. scaling law in a Llama-derived pretraining setup using the OpenLM library [21] and the RefinedWeb dataset [41] (Figure 1a). Our first observation is that accounting for the computational cost of the decoding layer (as done in Hoffmann et al. [25] but not in Kaplan et al. [30]) shifts compute-optimal scaling toward a more constant token-to-parameter ratio (Figure 1b). Second, we note that the constant-length warmup period of Kaplan et al. [30] is too long for smaller models, inflating the optimal number of tokens at lower compute budgets; scaling the warmup period with the model size further shifts the scaling in the Hoffmann et al. direction (Figure 1c). Next, we match the learning rate decay to the token budget of each configuration we test (as Hoffmann et al. [25] conjecture to be essential) but observe little effect on the compute-optimal scaling law (Figure 1d). Finally, we set the learning rate, batch size, and the AdamW $\beta_2$ parameters individually for each model size, leading to compute-optimal scaling that agrees closely with Hoffmann et al. (Figure 1e). Notably, the latter configuration uses a constant learning rate schedule, showing that learning rate decay is not essential for the Hoffmann et al. scaling law to emerge. We repeat our experiment on the OpenWebText2 dataset [18], observing similar results despite performing hyperparameter tuning only on RefinedWeb.

We complement our main results with the following analyses:

1. In the last phase of our experiments (Figure 1e) we choose different hyperparameters for each model size. To do so, we conduct a hyperparameter sweep for small-scale models and use the results to fit power laws for the optimal batch size and learning rate as a function of model parameters. This approach is inspired by DeepSeek [15], and our hyperparameter scaling laws roughly agree. However, we observe that setting the AdamW $\beta_2$ parameter to be 0.95 is suboptimal at smaller batch sizes (128 and below), and increasing it allows establishing clear trends from our small-scale hyperparameter sweep.

2. We study the scaling of the optimal loss as a function of the compute budget. We show that the steps we take to settle the Kaplan et al./Hoffmann et al. discrepancy (namely shortening warmup and scaling learning rate and batch size) significantly decrease this loss at smaller scales, but only marginally improve it at larger scales. In contrast, introducing a cosine learning rate decay

schedule substantially decreases the loss, with benefits persisting at larger scales. Similar to Hoffmann et al. [25], we observe some curvature on the optimal loss curve. Nevertheless, the optimal loss with tuned hyperparameters is fairly consistent with a saturating power law.

3. We calculate the computational cost of each of our experiments and plot how prediction quality improves as we consider larger training runs. We observe that the cost of our hyperparameter sweep is comparable to that of a scaling law fit experiment, but the compute saved by using a constant instead of a cosine learning rate schedule roughly makes up for that cost.

**Code and data release.** To facilitate future research, we share the data and the code necessary to reproduce our analyses and figures at `https://github.com/formll/resolving-scaling-law-discrepancies`. In addition, checkpoints of the models we train are available at `https://huggingface.co/formll/resolving-scaling-law-discrepancies`.

## 2 Preliminaries and experiment design

### 2.1 Notation and problem setting

We train language models of *size* $N$ on $D$ tokens of data (essentially without repetition). The precise definition of $N$ plays an important role in this paper: Unless mentioned otherwise, $N$ denotes the number of parameters in all the linear layers of the model. That is, $N$ excludes embedding layers, but includes the model's *head*: the final linear layer producing the predicted token logits. (In the models we train there is no tying of the embeddings and the head).

Let $\mathsf{FLOPs}(N, D)$ be the ammount of floating point operations (FLOPs) required to train a model of size $N$ on $D$ tokens. Throughout, we employ the approximation

$$\mathsf{FLOPs}(N, D) \approx 6ND \tag{1}$$

In Sections 3.1 and 3.2 we compare our definition of $N$ to the one used in Kaplan et al. [30]. In Appendix B we also discuss the effect of taking attention FLOPs into account and FLOP estimation approaches in other works.

Let $L(N, D)$ be the log loss (in expectation over the training data distribution and any randomness of the training procedure) obtained by a model of size $N$ trained for $D$ tokens.[4] Assuming a fixed compute budget $C$, we aim to predict

$$N^{\star}(C) \coloneqq \operatorname*{argmin}_{N > 0} L\left(N, \frac{C}{6N}\right) \approx \operatorname*{argmin}_{N > 0} \min_{D:\mathsf{FLOPs}(N,D)=C} L(N, D), \tag{2}$$

i.e., the model size yielding the smallest loss when trained with compute budget $C$ under the approximation (1).

We also let

$$D^{\star}(C) \coloneqq \frac{C}{6N^{\star}} \ \text{ and } \ \rho^{\star}(C) \coloneqq \frac{D^{\star}(C)}{N^{\star}(C)} = \frac{C}{6[N^{\star}(C)]^2} \tag{3}$$

denote the optimal number of tokens and the optimal token-to-parameter ratio. To predict these quantities, we use power laws of the form:

$$N^{\star}(C) \approx N_0^{\star} \cdot C^a \ , \ D^{\star}(C) \approx D_0^{\star} \cdot C^b \ \text{ and } \ \rho^{\star}(C) \approx \rho_0^{\star} \cdot C^r, \tag{4}$$

and fit the *exponents* $a, b, r$ and *coefficients* where $N_0^{\star}, D_0^{\star}, \rho_0^{\star}$ from data as described below.

### 2.2 Training setup

We train decoder-only Transformer language models using OpenLM [21], which integrates many of the architecture and training advances in Llama [55, 56] and subsequent works. We largely base our initial training configuration on the hyperparameter search in Gadre et al. [17]. Our setup does not replicate Kaplan et al. [30], but we match or closely approximate several key hyperparameters as discussed in Section 3. See Appendix C for a detailed description of our setup and chosen hyperparameters.

---

[4]This notation abstracts away the fact that there are many different models of size $N$ and many different ways to train them for $D$ tokens. Ideally, $L(N, D)$ represents the loss attained by the optimal architecture of size $N$ trained with the best possible training method that uses $D$ tokens. In practice, for any value of $(N, D)$ we consider only a single configuration, but this configuration is the result of architecture search and optimizer tuning, performed either directly or indirectly by building on prior work.

**Model set.** We search for compute-optimal models over a set consisting of 16 models with sizes ranging from 5M to 901M. We pick model layer numbers $l$ and widths $d$ such that $N$ increases by multiples of roughly $\sqrt{2}$ while the aspect ratio $d/l$ stays between 32 and 64 as suggested in Kaplan et al. [30]. The number of attention heads in each configuration is 4, as preliminary experiments showed this is optimal for smaller models, and increasing it did not noticeably improve larger models. Table 2 in the appendix specifies all the models in our grid.

**Data.** We perform our experiments on OpenWebText2 [18] which contains roughly 30B tokens of data from Reddit and resembles the WebText2 dataset used in Kaplan et al. [30], as well a RefinedWeb [41] dataset which contains roughly 600B tokens from CommonCrawl [1] and resembles the MassiveWeb dataset that formed roughly half of the data mix in Hoffmann et al. [25].

**Evaluation and FLOP grid.** We evaluate models on 160M tokens held out from the training data. We perform the evaluation whenever the product of $6N$ and the number $D$ of training tokens seen so far crosses an element of a *FLOP grid* of the form $\{1.25e16 \cdot 2^i\}_{i=0}^{11}$. This grid plays a central role in our data analysis. We also record the average training loss every 20 steps.

## 2.3 Data analysis

Our technique for estimating the compute-optimal power law is akin to the second (IsoFLOP-based) approach of Hoffmann et al. [25], but differs in several details. The approach consists of two steps: directly estimating $N^\star(C_i)$ for all $C_i$ in our FLOPs grid, and fitting a power law to these estimates. We briefly outline each step below and provide full details in Appendix D.

**Estimating $N^\star(C_i)$.** For each value of $C_i$, we train several models from our set (Table 2) for $C_i$ FLOPs and extract an *IsoFLOP curve* of loss vs. model size (see Figure 10). For FLOP values where validation loss is not available (specifically Section 3.1 and Appendix B) we use the smoothed training loss instead. We estimate $N^\star(C_i)$ and its uncertainty using a noise-and-interpolate procedure based on Gaussian noise with empirically-calibrated magnitude and Akima interpoaltion [3]. For every $C_i$, this yields a "bootstrap sample" population optimal size estimates; we take their median as the point estimate for $N^\star(C_i)$. The procedure also yields an estimate of the log-scale standard deviation of $N^\star(C_i)$ (shown as error bars in Figure 1).

**Fitting a power law.** We fit power laws of the form (4) by performing weighted linear regression in log space, with the weights inversely proportional to the squared log-space standard deviations computed above (i.e., log-space Gaussian maximum likelihood estimation). To obtain a point estimate for the power law parameters we fit the point estimates for each $N^\star(C_i)$ value. To quantify uncertainty, we fit power laws to bootstrap samples, obtaining a population of $N_0^\star$, $a$, and $N^\star(\cdot)$ samples. We construct confidence intervals from their quantiles.

## 3 Main results: settling the scaling law discrepancy

In this section, we describe in detail our main results, visualized in Figure 1, tabulated in Table 1 and plotted in detail in Appendix E. The following subsections address each panel of Figure 1 in order.

### 3.1 Reproducing the Kaplan et al. scaling law

To reproduce the Kaplan et al. scaling law, we match the setup of [30] in terms of the batch size ($2^{19}$ tokens) and in terms of the learning rate schedule (warmup for $3000 \cdot 2^{19} \approx 1.57$B tokens followed by cosine decay to zero at $2.5e5 \cdot 2^{19} \approx 131$B tokens). Other configurations do not match exactly, but the suite of models we train covers a range of sizes and compute similar to Kaplan et al. [30]. For this reproduction only, we also take the "model size" $N$ to be the number of parameters in all linear layers except the head (last decoding layer). That is, for a model of width $d$ and vocabulary size $v$, we subtract $d \cdot v$ from our usual definition of $N$ (see Table 2, last column).

As Figure 1a shows, with this setting we obtain a compute-optimal exponent $a$ and power law fits close to the power law $1.6e9(C/8.64e19)^{0.88}$ obtained by Kaplan et al. [30, Figure 14, left].

Table 1: Summary of our main results (described in Section 3).

| Experiment | Dataset | $a$ estimate | $R^2$ of $a$ fit | $\rho^\star$ range |
|---|---|---|---|---|
| Hoffmann et al. [25] | MassiveText | 0.5 | | |
| Kaplan et al. [30] | WebText2 | 0.88 | | |
| Adjusted Kaplan et al. [30] | WebText2 | 0.73 | | |
| Reproducing Kaplan et al. (§3.1) | OpenWebText2 | 0.864 (0.82, 0.90) | 0.998 | (5, 2617) |
| | RefinedWeb | 0.835 (0.82, 0.85) | 0.999 | (8, 1536) |
| Counting last layer FLOPs (§3.2) | OpenWebText2 | 0.699 (0.66, 0.72) | 0.998 | (8, 262) |
| | RefinedWeb | 0.706 (0.69, 0.72) | 0.998 | (9, 232) |
| Correcting warmup (§3.3) | OpenWebText2 | 0.603 (0.57, 0.63) | 0.994 | (7, 55) |
| | RefinedWeb | 0.602 (0.59, 0.62) | 0.993 | (7, 50) |
| Cosine decay (§3.4) | OpenWebText2 | 0.574 (0.54, 0.61) | 0.999 | (7, 42) |
| | RefinedWeb | 0.571 (0.56, 0.59) | 0.998 | (10, 39) |
| Optimizer tuning (no decay) (§3.5) | OpenWebText2 | 0.518 (0.49, 0.54) | 0.998 | (11, 22) |
| | RefinedWeb | 0.497 (0.49, 0.50) | 0.997 | (14, 16) |
| Reprod. adjusted Kaplan et al. (§H) | RefinedWeb | 0.717 (0.71, 0.72) | 0.992 | (12, 345) |

## 3.2 Counting last layer FLOPs

Kaplan et al. [30] chose to define model size without counting embedding parameters since they found this makes scaling laws in the infinite-compute regime more consistent across network depths [30, Figure 6]. Perhaps because their model head and embeddings had tied weights, this led them to also discount the contribution of the model head to the model's FLOPs per token [30, Table 1, last row]. However, as Table 2 reveals, not accounting for the model head leads to under-approximation that grows smoothly as model size decreases, from roughly $10\%$ at larger models to roughly $90\%$ at smaller models. Thus, counting the head FLOPs (i.e., using our definition of $N$) results in a significantly more accurate approximation. As shown in Figure 1b, switching to our model size count also reduces the exponent $a$ by more than $0.1$, closer to Hoffmann et al. but not all the way there.

## 3.3 Correcting learning rate warmup

Next, we address the duration of the learning rate warmup period, which Kaplan et al. [30] set proportionally to their full training duration, designed to reach complete convergence. Figure 2 (left) shows this warmup period is too long: for smaller-scale models, the optimal number of tokens as a function of compute is less than or close to the number of warmup tokens, and therefore these models are suboptimally trained. The same issue is evident in Figure 14 (right) of Kaplan et al. [30] which shows that for many compute budgets the optimal number of steps is below or close to the number of warmup steps (fixed at 3000). Figure 2 (left) also provides an intuitive explanation for the increased value of $a$: at smaller compute scales, models are 'forced' to use more training tokens than would otherwise be optimal in order to 'escape' the long warmup period. Having escaped, the warmup Once this warmup period is escaped, the optimal number of tokens grows only slowly, leading to a fast rate of increase in the optimal model size and hence the large exponent.

With the problem identified, we propose a simple heuristic for more appropriately choosing the warmup duration: for each model, we set the number of warmup tokens to be identical to the model size $N$. We validate our warmup heuristic in Appendix F. The bottom row of Figure 2b illustrates the validity of our new choice of warmup, showing that the optimal number of tokens is always at least 5 times greater than the (interpolated) duration of the warmup period corresponding to the model of the appropriate size. As is evident from this figure and from Figure 1c, shortening the warmup shifts the scaling law in the direction of Hoffmann et al. further, yielding an exponent $a$ of roughly 0.6.

## 3.4 Learning rate decay has limited impact on compute-optimal allocation

With learning rate warmup corrected, we turn to study learning rate decay, which Hoffmann et al. [25] conjecture to be a main cause of the difference between their result and Kaplan et al. [30]. We

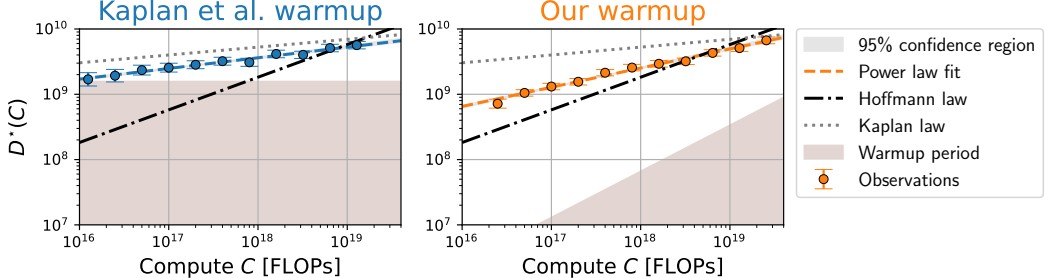

Figure 2: The optimal number of tokens $D^\star$ as a function of the compute budget $C$. **Left:** Using the warmup period of Kaplan et al. [30], smaller models reach compute-optimality during warmup. **Right:** Setting the number of warmup tokens to be identical to the model size (visualized using the power law fit) ensures models reach compute-optimality well after the warmup and yields a scaling law closer to Hoffmann et al.. We replicate these plots for all of our experiments in Appendix E.

observe that the long 131B tokens decay period in Kaplan et al. [30], which is aimed toward training to full convergence, means that their compute-constrained experiments see virtually no learning rate decay: Figure 2 shows that, at our compute scales, it is never optimal to train for more than 10B, which corresponds to less than $1.5\%$ decay with a cosine schedule.

To correct this, we follow the second approach of Hoffmann et al. [25] and choose the learning rate schedule for every model and FLOP budget individually. For each FLOP value in our grid, we pick the 7 models from Table 2 which yield token-to-parameter ratios in the range 1 to 100, and train them with a cosine learning rate schedule that decays to $1\%$ of the maximum learning rate when reaching the target FLOP value.[5] This is roughly twice as expensive as previous experiments, which required only a single training run for each model size (see additional discussion in Section 4.2). As Figure 1d shows, adding cosine decay results in a slightly cleaner linear trend ($R^2$ improves from 0.993 to 0.998) and an exponent slightly closer to the Hoffmann et al. scaling law (0.57 instead of 0.6), but most of the gap remains. Therefore, even with the FLOP count and warmup issues corrected, adding learning rate decay is not sufficient to reproduce the Hoffmann et al. scaling law.

### 3.5 Correcting batch size, learning rate and $\beta_2$

A final factor contributing to the Kaplan et al./Hoffmann et al. discrepancy is the choice of optimization hyperparameters, particularly the batch size: with a fixed batch size of $2^{19}$ tokens, compute-optimal models at smaller scales train for only a few hundred steps, which is likely too little. Kaplan et al. [30] notice this issue, and attempt to correct for it using post-processing based on an empirical model of large-batch size training [36]; we return to their result at the end of this section.

Here, we take the more direct approach of predicting near-optimal hyperparameters for each model size.[6] Since changing the batch size often also requires re-tuning the learning rate [19, 36, 49, 61], we sweep over both parameters for models of sizes 5M to 108M, with an additional validation sweep over models of size 220M. Initially, we kept $\beta_2$ at its previous value of 0.95. However, this led to poor results at smaller batch sizes: as the batch size gets smaller, the squared gradients become noisier, and AdamW requires more smoothing to obtain a correct denominator. Therefore, we added 0.99 and 0.999 to the sweep, obtaining improved performance on small batch sizes. This empirical observation matches the theoretical work by Zhang et al. [63], that showed that when the batch size is small, higher values of $\beta_2$ are crucial for the convergence of Adam. In Appendix G we describe the parameter sweep in full, validate the optimality of our prescribed hyperparameters on the largest model we train, and provide additional discussion about the role of $\beta_2$.

Figure 3 plots our estimates for the optimal values of batch size and learning rate for each model size. It shows clear trends, to which we fit power laws in the number of parameters $N$. Observing good

---

[5]We set the warmup period to be the minimum of the model size $N$ and $20\%$ of the total token budget. We ablate the final learning rate in Appendix F.

[6]More specifically, we predict the optimal hyperparameters per model size when trained for 20 tokens per parameter. As we discuss in Section 5.2, this choice of training budget is potentially an issue but further analysis in Appendix G.4 suggests it does not significantly impact our results.

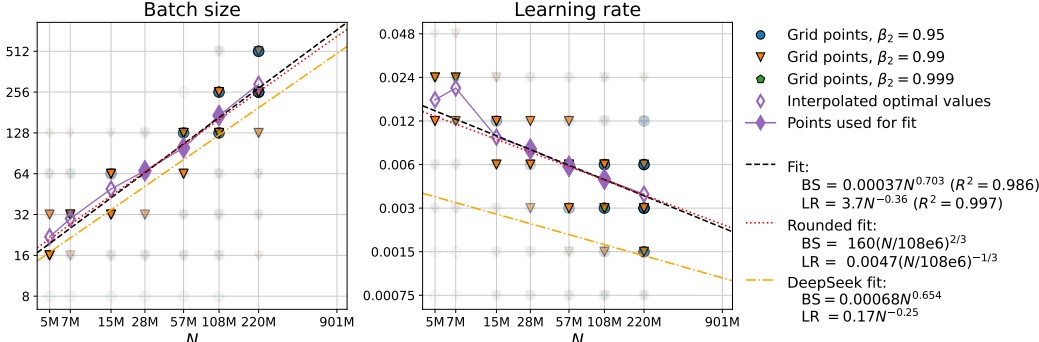

Figure 3: Fitting scaling laws for the optimal batch size and learning rate as a function of the model size $N$. Markers indicating grid points are shaded by their excess loss compared to all configurations for this parameter, reaching maximum transparency for loss that is suboptimal by $0.03$ or more. We also plot interpolation-based estimates of the optimal parameter values and fit them with power laws.

extrapolation to nearby values of $N$, we apply these power laws (with slight rounding) to select the batch size and learning rate for all model sizes and tabulate the results in Table 4. In Appendix G.5 we validate our learning rate and batch size scaling law for a model with 901M parameters.

Our parameter tuning approach is inspired by DeepSeek [15], who predict the optimal batch size and learning rate as a function of compute. Translating compute to model size using the Hoffmann et al. scaling law, we find remarkable agreement in the batch size predictions (a difference of less than $0.05$ in exponent and less than $60\%$ in predictions over our models), and somewhat different learning rate predictions (a difference of $0.11$ exponent and a factor of 2–3 in predictions), potentially due to using different weight decay. Both our results appear to contradict the conventional wisdom about the existence of a critical batch size [49, 61, 36] below which every batch size is good, finding instead an optimal batch size below which performance degrades. This suggests further tuning of $\beta_2$ or other hyperparameters may be warranted. We discuss DeepSeek [15] further in Section 5.1.

With the new hyperparameters, we obtain a close reproduction of the Hoffmann et al. scaling law (Figure 1e) with the scaling exponent matching $0.5$ to within $0.6\%$ and the predicted model size at Chinchilla compute within $15\%$ of Chinchilla's size. Notably, here we use a *constant learning rate schedule*, demonstrating that careful learning rate decay is not necessary for this scaling law to hold.

Finally, we reproduce the adjusted scaling law $N^\star(C) = 1.3\mathrm{e}9(C/8.64\mathrm{e}19)^{0.73}$ which Kaplan et al. [30] obtain by estimating the compute required to reach the same results at a sufficiently low batch size. To do so, we use our tuned hyperparameters as a proxy for suitable batch size and revert our previous corrections (head FLOP count and warmup duration). We obtain an exponent of $0.717$ and good agreement with their adjusted scaling law; see Figure 18 in Appendix H.

## 4 Additional Analysis

### 4.1 Trends in compute-optimal loss

Figure 4 shows the minimum loss achievable for each compute budget $C$ in the experiments shown in Figure 1. We estimate the minimum loss using the same interpolation procedure we use to extract the optimal parameter number $N^\star$ and token count $D^\star$. The figure shows that, at low compute scales, shortening the warmup duration and tuning hyperparameters leads to substantial loss improvements (each by up to 0.5 nat per token). However, at larger scales these interventions do not significantly improve the loss. In contrast, learning rate decay becomes increasingly beneficial as compute grows, and appears to also improve the rate of decrease in the loss. Perhaps coincidentally, the effects of overestimating the optimal loss (due to long warmup and large batch size) seem to closely offset the effect of underestimating computational cost (by discounting the contribution from the model's head): the first and last curves in Figure 4 closely overlap.

Similarly to Hoffmann et al. [25] we observe a curvature in the optimal loss, while Kaplan et al. [30] report a near-perfect power law behavior. This difference is due to a combination of the difference in FLOP counts discussed in Section 3.2 and the fact that the experiments of Hoffmann et al. [25]

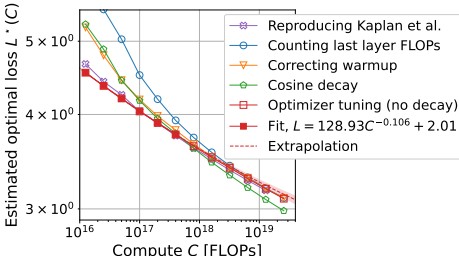
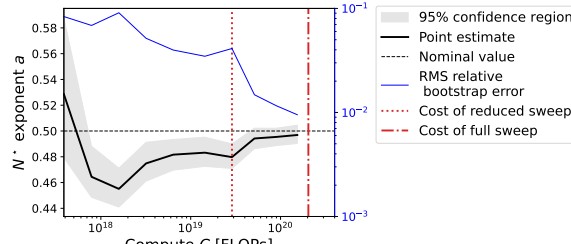

Figure 4: The minimum loss achievable by models with compute budget $C$. For the Kaplan et al. scaling law reproduction, we estimate $C$ as in Section 3.1. See expanded version in Figure 19.

Figure 5: Compute optimal exponent prediction, confidence, and root-mean-square relative error as a function of the total scaling experiment budget for the tuned optimizer experiment described in Section 3.5.

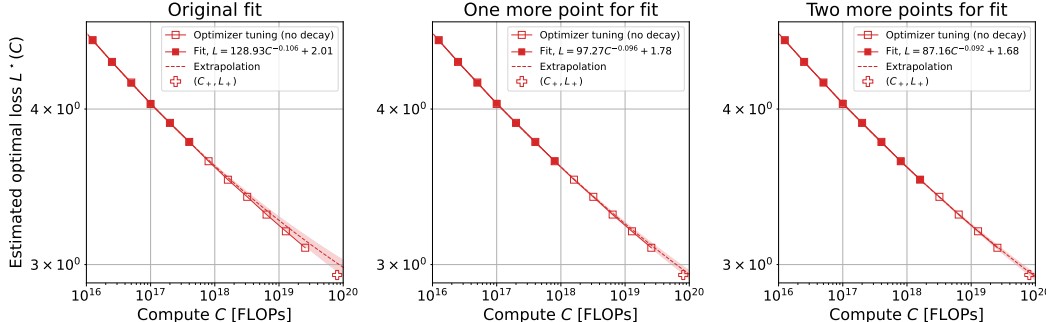

Figure 6: The compute-optimal loss curve of Figure 4 extended to compute budget $C_+ \approx 8\text{e}19$ by training a single model with size, learning rate and batch size determined using our scaling laws. Each subplot uses a different number of lower-compute loss measurement to fit the loss trend.

extend to higher compute budgets where the loss is closer to its irreducible level. Indeed, for the tuned optimizer experiment (Section 3.5) we find that a saturating power law fits the optimal loss and extrapolates well, while extrapolating poorly for other experiments (see Figure 19 in the appendix). This suggests that a predictable trend in $L(N^\star(C), D^\star(C))$ is an indicator of locally-optimal hyperparameters. The exponent of our saturating power fit is approximately $-0.1$, twice as large as the exponent found in Kaplan et al. [30].

Finally, we validate our compute-optimal loss scaling law by training and evaluating a model using a larger compute budget. Specifically, we train a 901M parameter model with a compute budget of $C_+ \approx 8\text{e}19$ FLOPs, which our scaling law predicts to be compute-optimal, using the batch size and learning prescribed by our hyperparameter scaling laws (see Table 4), reaching a loss value of $L_+ = 2.943$. In Figure 6, we add the point $(C_+, L_+)$ to the red curve in Figure 4 and find that it falls within the predicted trend. Notably, $L_+$ is obtained with a single training run using our predicted optimal configuration, while at lower compute values we estimate the optimal loss by interpolating an IsoFLOP curve.

## 4.2 Scaling law accuracy as a function of compute

We now estimate the computational cost of our scaling law experiments, quantifying the effect of the learning rate schedule, and plot how our predictions improve and become more confident with increased computation. We find that the training cost of each experiment that utilized a fixed learning rate schedule was $1.54\text{e}20$ FLOPs, while the experiments that used a varying-length cosine learning rate schedule required $2.99\text{e}20$ FLOPs; essentially double the compute; see Appendix J for more details. We also find that the cost of the hyperparameter sweep described in Section 3.5 was $2.04\text{e}20$ FLOPs—slightly less than the combined cost of two scaling experiments that leveraged it (one on each dataset). Moreover, in hindsight, we could have arrived at similar hyperparameters using only models of size at most 57M and a simple heuristic for choosing $\beta_2$ based on batch size, which would have cost only $1.44\text{e}19$ FLOPs.

Figure 5 shows the evolution of the predicted compute-optimal model size exponent $a$, its confidence interval, and a measure of the prediction accuracy as we modulate the experiment's compute budget by truncating our FLOP grid. The figure shows that the prediction becomes steadily more accurate and confident as compute increases. We present these results, as well as the results in Section 4.1, on the OpenWebText2 dataset as well (see Appendix I).

## 5 Discussion

### 5.1 Related work

While neural scaling laws precede the advent of large language models [24, 45], breakthroughs in model [57] and data [42, 44] scaling allowed Kaplan et al. [30] to demonstrate the dramatic possibility of unbounded improvement with scale, triggering an explosion in the literature on the topic. Here we focus on the relatively fewer works that tackle optimal resource allocation under a *compute* constraint.

For language modeling, Hu et al. [26] and DeepSeek [15] repeat subsets of the analyses in Hoffmann et al. [25] and derive compute optimal scaling laws. Employing Approach 3 of Hoffmann et al. [25] (see also [9]), Hu et al. [26] find that, for their models, optimal scaling favors larger token-to-parameter ratios than in Hoffmann et al. [25] and in our results. They attribute this difference to modeling improvements since [25] and argue the same holds for Llama 2 [56]. However, our setup incorporates most of the advances in Llama 2 and still produces power laws very close to Hoffmann et al. [25]. Like us, DeepSeek [15] perform hyperparameter tuning and use isoFLOP analysis to determine compute-optimal model sizes on multiple datasets. While they arrive at an exponent on the order of Hoffmann et al. [25] for the main dataset they study, they report a higher exponent $a = 0.578$ for OpenWebText2 (i.e., predicting lower token-parameter-ratio at scale), which they attribute to the superior quality of the dataset. We also study this dataset but arrive much closer to the Hoffmann et al. scaling law. We conjecture the larger exponent might be due to repeating training data, which likely occurred in their experiment given the dataset's limited size and their compute budget. Settling these discrepancies could be a source of further valuable lessons on optimal model scaling.

Recent work also studies compute-bounded scaling laws beyond the compute-optimal regime. Informed by the increasingly common practice of training medium-scale models beyond compute optimality [e.g., 55, 56, 29], Sardana and Frankle [46] account for the expected inference cost of the model, showing that it naturally skews optimal settings toward smaller models. Gadre et al. [17] directly predict the loss and downstream performance for models trained past the point of compute optimality, and Muennighoff et al. [37] model joint compute-data bottlenecks. All three works rely on the Hoffmann et al. law as a reference point, with [17, 37] baking it to their parametric forms.

Compute-optimal scaling is studied beyond the language domain, particularly in vision. Henighan et al. [23] study autoregressive modeling for a variety of tasks and find scaling laws roughly consistent with the Kaplan et al. [30] adjusted scaling law (with exponent $a = 0.73$). That work shares the methodological issues described in the top row of Figure 1 (FLOP count and long warmup), but performs hyperparameter tuning for smaller scale models; in Appendix H we reach similar results when doing the same. Zhai et al. [59] characterize the compute-efficient frontier of Vision Transformers (ViTs), while Cherti et al. [13] studies compute constrained scaling of CLIP models. However, they do not offer a power law for scaling model size with compute. Alabdulmohsin et al. [4] tackle model design under an *inference compute* constraint by fitting multi-term parametric forms to obtain predictions for the optimal ViT shape. Goyal et al. [20] point out an intricate interplay between data filtering and compute constraints. Finally, Bachmann et al. [7] study compute-optimal scaling of MLP's and obtain exponent $a = 0.35$, suggesting that MLP require much more rapid data growth than more sophisticated architecture. Overall, whether and to what extent does Hoffmann et al. scaling hold in the vision domain remains a compelling open problem.

Recent theoretical work study compute-optimal scaling laws in simplified, analytically tractable settings [11, 33, 38, 28]. In particular, Paquette et al. [38] obtain a power law with exponent $a = 0.5$ (as in Hoffmann et al. [25]) for a random-feature linear regression setting [35, 5], conjecturing that this is a part of a broader, universal phenomenon. Jeon and Van Roy [28] also establish an exponent of 0.5 for data generated by infinite-width two-layer ReLU networks, using information-theoretic arguments.

We also remark on two themes of our paper that draw from prior work. The first is the importance of hyperparameter tuning: several works [30, 27, 17, 15] make the case that smooth, predictable

scaling laws emerge when models on all scales are properly tuned. Our work (and particularly Section 4.1) provides another example of this principle and agrees with previous observations that tuning is particularly important at smaller scales. Second, previous studies [59, 8, 15, 26] as well as the concurrent work [22], propose alternative learning rate schedules that address a key shortcoming of cosine decay: the need to commit to a step budget in advance. We consider a constant learning rate that requires no commitment at all. We show this simple choice suffices to reproduce the Hoffmann et al. law and quantify the computational savings compared to a cosine schedule. However, Section 4.1 (and also [25], among others) show that in terms of loss, the constant schedule clearly underperforms the cosine schedule.

**Concurrent and independent work.** Pearce and Song [40] also study the discrepency between the Kaplan et al. and Hoffmann et al. scaling laws. By re-analyzing data extracted from the Hoffmann et al. [25] experiments by Besiroglu et al. [9], they identify the last layer FLOP count as a cause for the discrepancy. Moreover, they report on a small-scale experimental study (with model sizes up to 5M and training tokens number up to 530M) in which they observe that a non-decaying learning rate schedule is sufficient for reproducing the Hoffmann et al. exponent and that learning rate tuning is necessary. These results independently corroborate part of our observations in Sections 3.2, 3.4 and 3.5. Pearce and Song [40] do not identify the warmup duration issue we describe in Section 3.3. As a consequence, when reproducing the Kaplan et al. exponent they reach a value close to 0.73 rather than the 'raw' value 0.88 reported in Kaplan et al. [30] (see discussion in Section 3.1, Section 3.5, and Appendix H). In addition, our experiments roughly match the Kaplan et al. [30] compute budget, which is about 3 orders of magnitudes larger than budget in Pearce and Song [40], and we perform careful tuning of both the learning rate and the batch size.

## 5.2 Limitations

Computational scale is a notable limitation, as well as a defining feature, of our results: our experiments are roughly on the scale of those in Kaplan et al. [30] but are substantially smaller than those of Hoffmann et al. [25]. Scaling may effectively mitigate each of the issues we identify: with scale, the contribution of the model head becomes negligible, any (fixed) warmup period eventually becomes reasonably long, and hyperparameter sensitivity decreases, as shown in Figure 4 and Figure 15. Nevertheless, we believe that experimental protocols that induce correct scaling behavior at low computational budgets are crucial for developing the empirical science of machine learning, particularly in academic settings.

Due to limited compute budgets, our hyperparameter sweep only targeted the smaller models in our grid, and furthermore trained each model for only $20N$ steps, i.e., the optimal point according to the Hoffmann et al. scaling law. This raises the concern that the hyperparameters we chose unfairly favor models trained for that particular token-to-parameter ratio, and rerunning our experiment with perfect tuning for each model size *and* each token-to-parameter ratio would have yielded different results. We believe this is unlikely: at small scales (where hyperparameter tuning is crucial) our original set of hyperparameters favored higher token-to-parameter ratios because they still had a sufficient number of steps to train for, and therefore choosing hyperparameters specifically for them is not likely to result in significant gains. In Appendix G.4 we analyze our existing tuning results to estimate the potential gains from perfect tuning, and find that they are likely to have small impact on our conclusions. Moreover, transferring our hyperparameters to another dataset yields similar results.

Finally, a broader limitation of compute-optimal scaling as defined by Kaplan et al. [30], Hoffmann et al. [25] and our work, is that it only concerns the pretraining loss rather than more direct measures of a model's capabilities. Here again, scale is an issue: most zero-shot and in-context capabilities do not emerge at the scales we consider here, and predicting them from small-scale proxies is an important open problem [48, 17]. Instead, it is possible to study downstream performance via fine-tuning, though this may cause the clean scaling patterns seen in pretraining to break down [54], potentially because the fine-tuning procedure is sensitive to the choice of hyperparameters [27].

## Acknowledgments

We thank Georgios Smyrnis, Samir Yitzhak Gadre, Achal Dave, and Mehdi Cherti for helpful discussion and assistance with OpenLM and with the JSC cluster.

TP and YC acknowledge support from the Israeli Science Foundation (ISF) grant no. 2486/21 and the Adelis Foundation. MW was supported in part by a Google Fellowship. JJ acknowledges funding by the Federal Ministry of Education and Research of Germany under grant no. 01IS22094B WestAI - AI Service Center West. LW acknowledges funding from Open Philanthropy. We gratefully acknowledge compute budget granted by Gauss Centre for Supercomputing e.V. and by the John von Neumann Institute for Computing (NIC) on the supercomputers JUWELS Booster and JURECA at Jülich Supercomputing Centre (JSC).

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

# A   Interpreting Hoffmann et al.'s hypothesis

Hoffmann et al. [25] hypothesize that the scaling law discrepancy is due to a difference in learning rate schedules. The term "learning rate schedule" comprises both the warmup and decay components, and it is plausible to interpret the "Modeling the scaling behavior" paragraph in Hoffmann et al. [25] §2 as conjecturing that both components contribute to the scaling law discrepancy. However, we believe that Hoffmann et al. emphasize the decay component of the learning rate schedule in their hypothesis due to two main reasons.

1. *Figure A1 in Hoffmann et al.* In §2, Hoffmann et al. refer to Figure A1 to provide evidence that the learning rate schedule affects compute optimal scaling. However, in this figure they only vary the decay component of the schedule, keeping the warmup fixed. Hence, it appears that—in the scope of their hypothesis—Hoffmann et al. equate learning rate schedule with learning rate decay.

2. *Subsequent literature.* In Hu et al. [26] §4.1, Hägele et al. [22] §1 and in [2] the authors refer to the decay component of the learning rate schedule as the cause of the discrepancy. This suggests that the community has interpreted Hoffmann et al.'s hypothesis as referring to the decay component.

# B   Estimating FLOPs and accounting for attention

In this section, we compare our definition of model size $N$ to three alternatives and also discuss choices made by related work. Before that, we provide the precise expression for computing $N$ (by our definition) from the model depth $l$, width $d$, and vocabulary size $v = 50\,432$. Due to efficient implementation considerations, OpenLM sets the model's feedforward dimension to $d_{\text{FF}} = 256 \left\lfloor \frac{255 + \lfloor 8d/3 \rfloor}{256} \right\rfloor$. Since each SwiGLU feedforward block has 3 $d_{\text{ff}} \times d$ parameter matrices, and since each attention block has $4d^2$ parameters in linear layers, our total estimate is:

$$N = (3d_{\text{FF}} + 4d)dl + dv. \tag{5}$$

We begin by considering, instead of the number of weights in linear layers, the total number of non-embedding learnable weights $N_{\text{exact}}$ (e.g., including also LayerNorm gains). The fourth column of Table 2 shows that the difference between this number and $N$ is negligible. We also note that embedding layers have a negligible contribution to the model FLOP counts, since they do not require matrix-vector products.

Consequently, the only non-negligible source of error in the approximation $\text{FLOPs}(N, D) = 6ND$ is the attention layers. Since in OpenLM the attention dimension is identical to the model width, Kaplan et al. [30, Table 1] shows that the attention operation costs an additional $6nd$ FLOPs per token per layer for a forward and backward pass, where $n = 2048$ is the sequence length. Thus, if we define an *effective* model size of

$$N_{\text{eff}} \coloneqq N + ndl, \tag{6}$$

we have that $6N_{\text{eff}}D$ captures the cost of training the model for $D$ tokens, including attention.

We now consider the difference between these approximations and its effect on compute-optimal scaling laws. The fifth column of Table 2 compares $N_{\text{eff}}$ to $N$. It shows that the ratio $N_{\text{eff}}/N$ changes smoothly between roughly 1.1 to roughly 1.2 and back to 1.1 as our model sizes grow. We note that had this ratio been completely constant, there would have been no essentially no difference between working with $N$ and working with $N_{\text{eff}}$ since a power law in one would correspond directly to a power law in the other. Since in our model grid this ratio is approximately constant, we expect to see limited differences between the scaling laws resulting from each definition. Figure 7 confirms this expectation, showing quantitatively and qualitatively similar results to Figure 1. Consequently, we cannot determine with certainty which definition is more appropriate for predicting compute-optimal model sizes. Nevertheless, we observe our final experiment (with parameter tuning) predicts that the optimal (effective) model size at the Chinchilla/Gopher compute scale to be about 16 B parameters larger than the one size predicted using our standard definition. These predictions are directly comparable since model size and effective model size are essentially identical at these scales. If we take the Hoffmann et al. scaling law as ground truth, then the prediction we get using $N_{\text{eff}}$ is a bit worse.

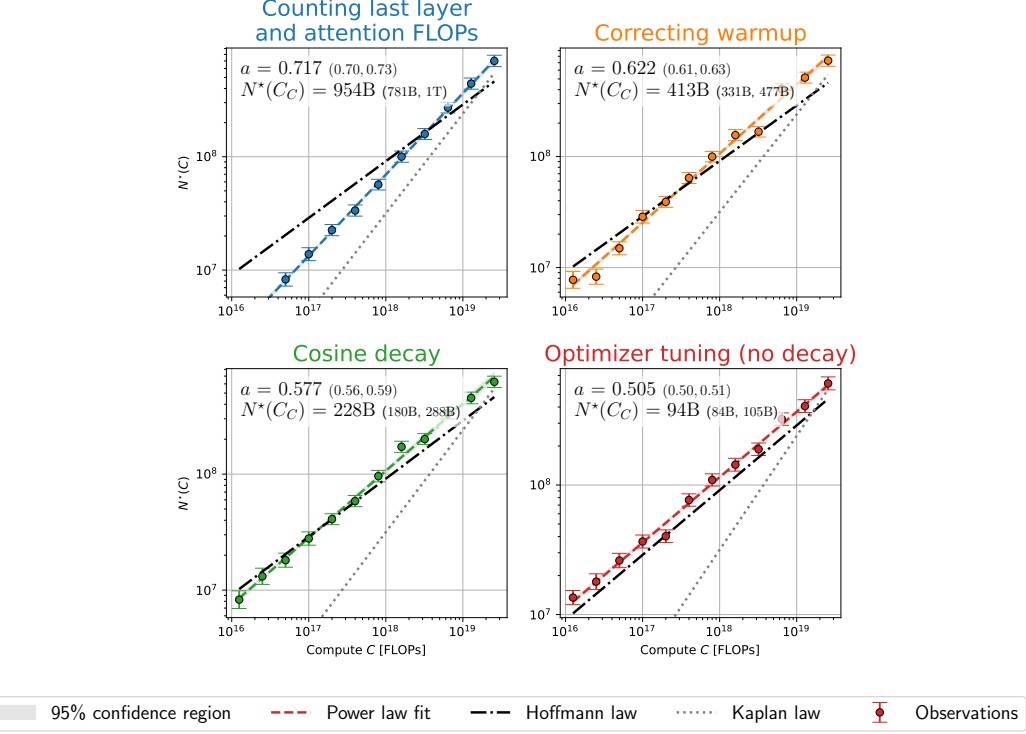

Figure 7: A reproduction of Figure 1 where we replace our standard definition of model size $N$ with the effective model size $N_{\text{eff}}$ (defined in eq. (6)) for which the approximation $6N_{\text{eff}}D$ also captures the FLOPs cost of attention operations in the model. We observe similar results to Figure 1, though our final experiment produces a mildly higher prediction for the optimal model size at the Chinchilla/Gopher compute scale.

Finally, we touch on a third measure of model size, which does not count the contribution of the model's head to the FLOP count. That is, we consider

$$N_{\text{Kaplan}} \coloneqq N - dv. \tag{7}$$

This is the definition that Kaplan et al. [30] use in their experiment, approximating the flop count as $6N_{\text{Kaplan}}D$. Their main motivation for this choice is an observation that not counting embedding parameters leads to more predictive scaling laws in the unlimited compute regime. However, as the final column of Table 2 shows, this approximation leads to a large, systematic error in FLOPs counts for smaller models. In Sections 3.1 and 3.2 we show that this is one of the primary factors behind the Kaplan et al./Hoffmann et al. discrepancy.

We conclude this section with an overview of the model size definitions used by related works other than Kaplan et al. [30]. Henighan et al. [23] use $N_{\text{Kaplan}}$ as Kaplan et al. [30] and observe high scaling exponents as a result. Hoffmann et al. [25] account for both linear and attention layers in their FLOP computation essentially using $N_{\text{eff}}$ in their first two estimation approaches. However, their third approach appears to ignore the attention FLOPs and also count the embeddings parameters, i.e., setting $N' = N + dv$. DeepSeek [15] compare 3 definitions of model size, including $N_{\text{Kaplan}}$, $N$, and a hybrid of $N_{\text{eff}}$ and $N_{\text{Kaplan}}$ that takes attention into account and ignores the model head. They report that the latter option gives the best prediction of the compute-optimal loss at large scales. However, we note that both [25] and [15] claim that attention costs double the FLOPs mentioned in Kaplan et al. [30]; we believe this is likely because they do not account for the fact that the attention is *causal*, meaning it requires only half the FLOPs of an unstructured matrix-vector product. Finally Muennighoff et al. [37], Gadre et al. [17], Hu et al. [26] use $N$ as we do.

Table 2: Model architectures and different parameter counts, in millions. The column $N$ gives our definition of models size, $N_{\mathrm{exact}}$ is the exact number of trainable parameters in the model, $N_{\mathrm{eff}}$ is the effective size which also accounts for the cost of attention operatoins, and $N_{\mathrm{Kaplan}}$ does not count parameters in the model's head. See Appendix B for precise definitions and discussion.

| Depth | Width | $N$ | $N_{\mathrm{exact}}$ | $N_{\mathrm{eff}}$ | $N_{\mathrm{Kaplan}}$ |
|---|---|---|---|---|---|
| 3 | 96 | 5.173 | 5.176 (+0.05%) | 5.763 (+11.4%) | 0.331 (-93.5%) |
| 4 | 128 | 7.504 | 7.508 (+0.06%) | 8.552 (+13.9%) | 1.049 (-86.0%) |
| 5 | 160 | 9.810 | 9.817 (+0.07%) | 11.45 (+16.7%) | 1.741 (-82.2%) |
| 6 | 224 | 15.60 | 15.61 (+0.07%) | 18.35 (+17.6%) | 4.301 (-72.4%) |
| 8 | 288 | 22.49 | 22.51 (+0.08%) | 27.21 (+20.9%) | 7.963 (-64.5%) |
| 9 | 320 | 28.67 | 28.70 (+0.08%) | 34.57 (+20.5%) | 12.53 (-56.2%) |
| 10 | 384 | 37.06 | 37.09 (+0.08%) | 44.92 (+21.2%) | 17.69 (-52.2%) |
| 12 | 480 | 57.38 | 57.43 (+0.08%) | 69.18 (+20.5%) | 33.18 (-42.1%) |
| 14 | 576 | 84.79 | 84.85 (+0.08%) | 101.3 (+19.4%) | 55.74 (-34.2%) |
| 15 | 640 | 108.5 | 108.5 (+0.07%) | 128.1 (+18.1%) | 76.19 (-29.7%) |
| 18 | 704 | 149.0 | 149.1 (+0.07%) | 175.0 (+17.4%) | 113.5 (-23.8%) |
| 21 | 832 | 220.9 | 221.0 (+0.06%) | 256.7 (+16.2%) | 178.9 (-19.0%) |
| 23 | 1024 | 347.1 | 347.3 (+0.05%) | 395.3 (+13.9%) | 295.4 (-14.8%) |
| 26 | 1120 | 455.3 | 455.5 (+0.05%) | 514.9 (+13.1%) | 398.8 (-12.4%) |
| 26 | 1312 | 612.0 | 612.2 (+0.05%) | 681.8 (+11.4%) | 545.8 (-10.8%) |
| 30 | 1504 | 901.7 | 902.1 (+0.04%) | 994.1 (+10.2%) | 825.9 (-8.41%) |

## C    Additional training setup description

**Modeling.**    We train decoder-only Transformer [57] language models for next-token prediction using OpenLM, a Pytorch [39] for efficient training of medium scale language models. We use the library with largely the same configuration as [17], leveraging xFormers [31], bfloat16 automatic mixed precision, (qk)-LayerNorm [6, 16, 58], SwiGLU [50], depth-scaled initialization [60], and rotary positional embeddings [53]. We use the GPT-NeoX-20B tokenizer [10] whose vocabulary size of 50432 closely matches the vocabulary size of Kaplan et al. [30]. We use a sequence length of 2048 which is twice the sequence length used in Kaplan et al. [30], but we attempt to match parameters like batch size and warmup duration in their size in tokens. We do not tie the weights of the embeddings and head layers.

**Optimization.**    Throughout the paper, we use the AdamW optimizer [34] to minimize the standard log loss with an additive z-loss term for stabilization [14] (coefficient 1e−4) as an auxillary loss (for our analysis, we record the log loss without the z-loss term in both train and validation). As advocated for in Wortsman et al. [58], we use independent weight decay [34] with parameter 1e−4, i.e., we set the "weight decay" parameter in the standard PyTorch AdamW implementation to be $1\mathrm{e}{-}4/\eta$, where $\eta$ is the base learning rate. In Table 3 and Table 4 we describe our choice of hyperparameter in our experiments.

**Hardware and computational cost.**    We train our models on a cluster with 40GB A100 GPU's, using between 4–32 GPU's in parallel per training run. We use the OpenLM/PyTorch distributed data parallel implementation as well as gradient checkpointing. According to our logs, the total compute cost of all the experiments going into this paper is 22.3K GPU hours., and the total FLOP count is 3.03e21 FLOPs.

**Data repetition.**    The datasets we work with are large enough to allow us to perform all of our training runs without any data repetition. However, due to two software issues, some experiments experienced limited data repetition. In particular, data going into our hyperparameter sweep might

Table 3: Fixed hyperparameters. We use these values in the experiments in Sections 3.1 to 3.4 and later tune part of them as described in Section 3.5 and Table 4 below.

| Name | Value |
| --- | --- |
| Batch size | 256 |
| Learning rate | 3e−3 |
| AdamW independent weight decay | 1e−4 |
| AdamW $\beta_1$ | 0.9 |
| AdamW $\beta_2$ | 0.95 |
| $Z$-loss weight | 1e−4 |

Table 4: Tuned hyperparameters. We choose these parameters according to the scaling law we present in Section 3.5. Due to parallelization requirements, we round batch size to be a multiple of number of GPUs used in each run. We also round learning rate to two significant digits. All other hyperparameters are as in Table 3.

| $N$ (millions) | Learning rate | Batch size | $\beta_2$ |
| --- | --- | --- | --- |
| 5 | 0.013 | 20 | 0.99 |
| 7 | 0.011 | 28 | 0.99 |
| 9 | 0.011 | 32 | 0.99 |
| 15 | 0.009 | 44 | 0.99 |
| 22 | 0.008 | 56 | 0.99 |
| 28 | 0.0074 | 64 | 0.99 |
| 37 | 0.0068 | 80 | 0.99 |
| 57 | 0.0059 | 104 | 0.99 |
| 84 | 0.0051 | 128 | 0.99 |
| 108 | 0.0047 | 160 | 0.99 |
| 149 | 0.0043 | 192 | 0.99 |
| 220 | 0.0038 | 256 | 0.95 |
| 347 | 0.0032 | 320 | 0.95 |
| 455 | 0.003 | 448 | 0.95 |
| 611 | 0.0027 | 512 | 0.95 |
| 901 | 0.0024 | 640 | 0.95 |

have been repeated up to 10 times. Moreover, on OpenWebText2, some of our larger-scale training runs might have seen data repeated up to 4 times. We believe this had limited to no impact on our results, as the hyperparameter sweep involved fairly small models unlikely to be able to memorize, while [37] show that 4 data repetitions have only a marginal effect on model performance. In our main experiments on the much larger RefinedWeb dataset we have verified that no data repetition occurred.

## D  Additional data analysis details

This section provides a comprehensive description of our procedure for fitting the power law for $N^\star(C)$. Our procedure for the $D^\star$ power law is analogous, using the relationship (3).

**Training loss smoothing.**  We smooth the training loss using a variable-length smoothing averaging window. In particular, we estimate the loss at step $i$ as the average of the losses in steps $i - \lfloor pi \rfloor$ to

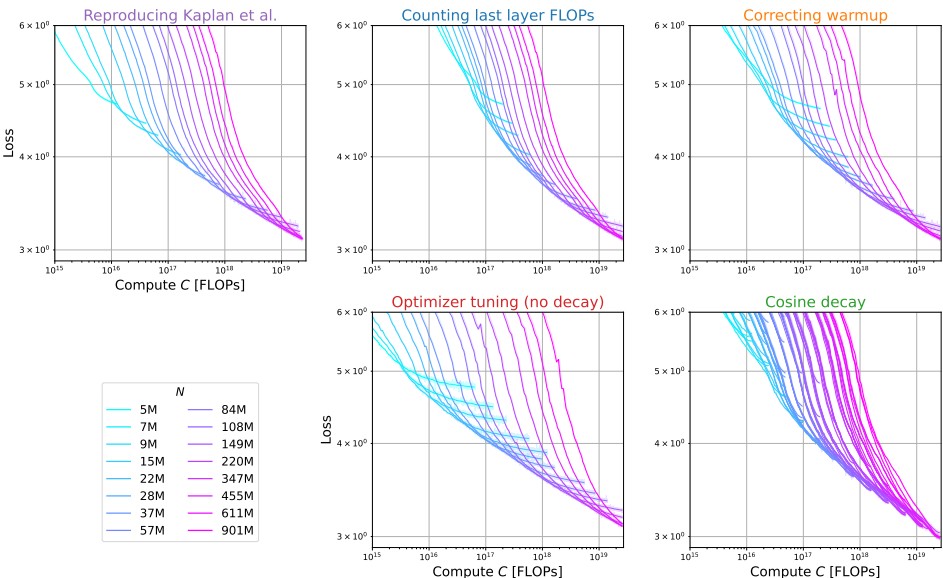

Figure 8: Training loss vs. compute for our main experiments on the RefinedWeb dataset. The smoothed loss is overlaid on semi-transparent raw loss.

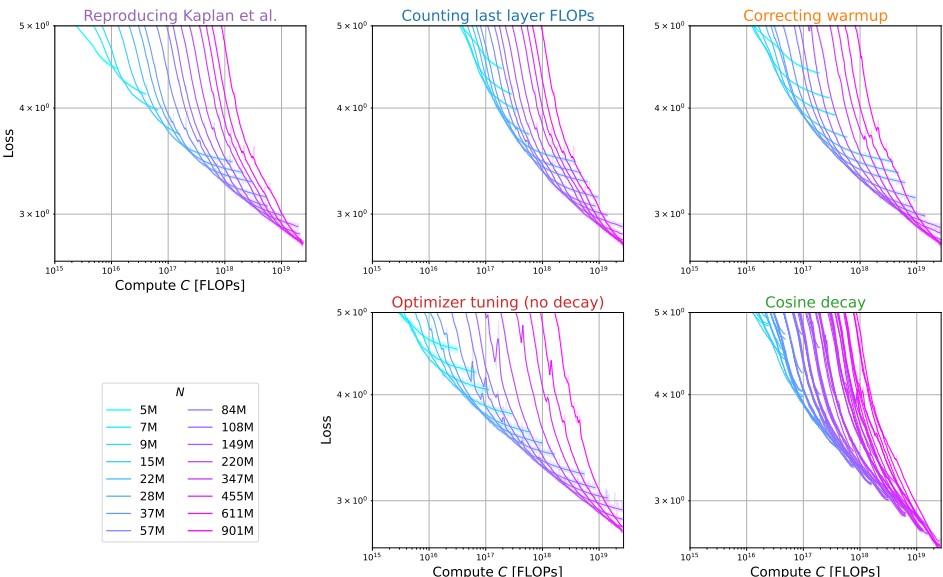

Figure 9: Training loss vs. compute for our main experiments on the OpenWebText2 dataset. The smoothed loss is overlaid on semi-transparent raw loss.

$i + \lfloor pi \rfloor$, for $p = 0.05$. We also compensate for the lag introduced by logging the averaged training loss every $k = 20$ by shifting the training loss's index $k/2$; this compensation is quite important for matching the validation loss early in the optimization. We have verified that the smoothed training loss matches the validation loss (where it is available) roughly to the validation loss's sampling error.

**Fetching the loss at $C_i$ FLOPs from a single run.** To estimate the loss of a model of size $N$ trained to compute $C_i$ we linearly interpolate the validation/training loss (in log-space) at the two steps closest to $C_i/(6NB)$, where $B$ is the batch size in tokens. We also require the nearest step to be within 10% of $C_i/(6NB)$, and do not return the loss if not such step exists. For most of our experiments, we compute the validation loss precisely at step $\lceil C_i/(6NB) \rceil$. However, for the

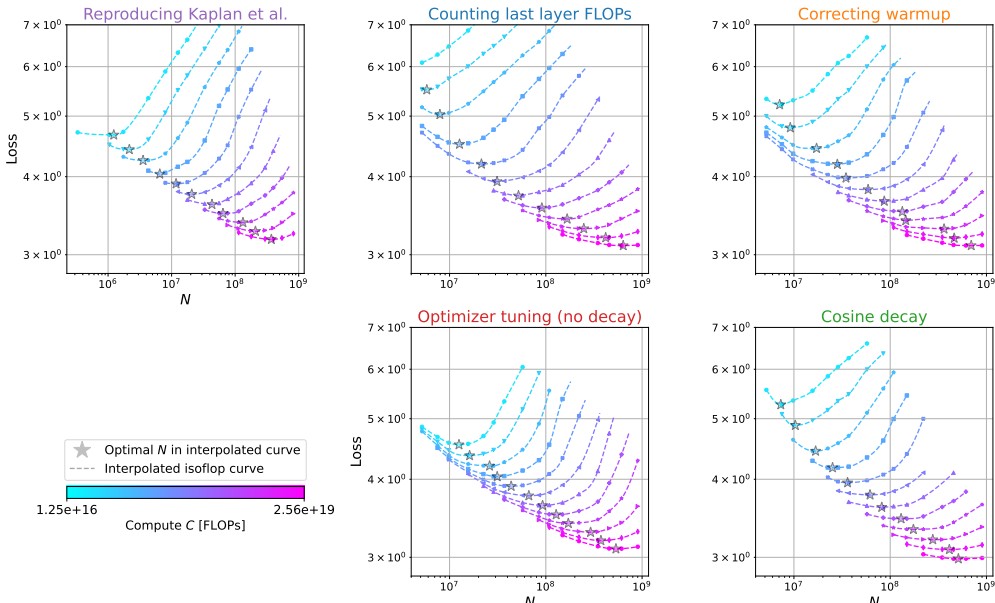

Figure 10: IsoFLOP curves for our main experiments on the RefinedWeb dataset. The values marked with stars are estimates of $N^\star(C_i)$ for the depicted $C_i$ values. See Appendix D for a related discussion.

experiments in Section 3.1 and Appendix B, which consider alternative definitions of $N$ we do not have validation loss samples, and we use the training loss instead.

**Estimating loss noise.** Defining the ideal loss as the population log loss in expectation over model training, there are two sources of error in estimating it: finite samples in the validation set, and variation between training seeds. We estimate the former directly by storing the validation loss on 100 subsamples of the holdout data and find the standard deviation to be in the range $0.001$–$0.002$ across the different experimental settings. To gauge the error due to seed variance, we train smaller-scale models from our grid on 7 seeds using the tuned hyperparameters for $20N$ tokens each. We find a roughly log-linear relationship (see Figure 12) between the (post-warmup) smoothed training loss and the inter-seed standard deviation. For RefinedWeb (Table 4), it appears to saturate around the sampling error, and we heuristically assign standard deviation $0.05$ to samples with loss $> 7$, standard deviation $0.002$ to samples with loss $< 3$, and linearly interpolate the standard deviation in log-space to samples with loss in the range $[3, 7]$. For OpenWebText2 we observe significantly more cross-seed variance as well as less stable loss during training (compare the loss curves in Figures 8 and 9), potentially due to a difference in document lengths. Therefore, we set our standard deviation estimate to go from $0.1$ at loss $6$ to $0.01$ at loss $3$, saturating outside the interval and log-space interpolating inside it.

**Estimating $N^\star(C_i)$ and its uncertainty.** Given a list of $N$ values and their respective loss samples at compute $C_i$ (fetched as described above), we estimate the optimal value $N^\star(C_i)$ using the following bootstrap-like procedure. For each bootstrap sample, we add independent Gaussian noise to each loss, whose standard deviation is determined according to the heuristic formula described above. We then interpolate the curve of loss vs. $N$ using Akima [3] interpolation in log-space and find the value minimizing the interpolant; this forms our population of bootstrap samples for $N^\star(C_i)$ estimate. We estimate their standard deviation in log-space, and take the maximum between that value and one-third of the average log-spacing on the grid (roughly $\frac{1}{3} \log \sqrt{2}$). Occasionally, $N^\star$ appears on the edge of the $N$ grid (though we attempt to avoid this in our experiment design). If more than half of the bootstrap samples land at the edge of the grid, we omit the value of $C_i$ from the subsequent power law fit. Otherwise, we keep only the samples outside the grid edge, and blow up the standard deviation estimate by the fraction of omitted samples.

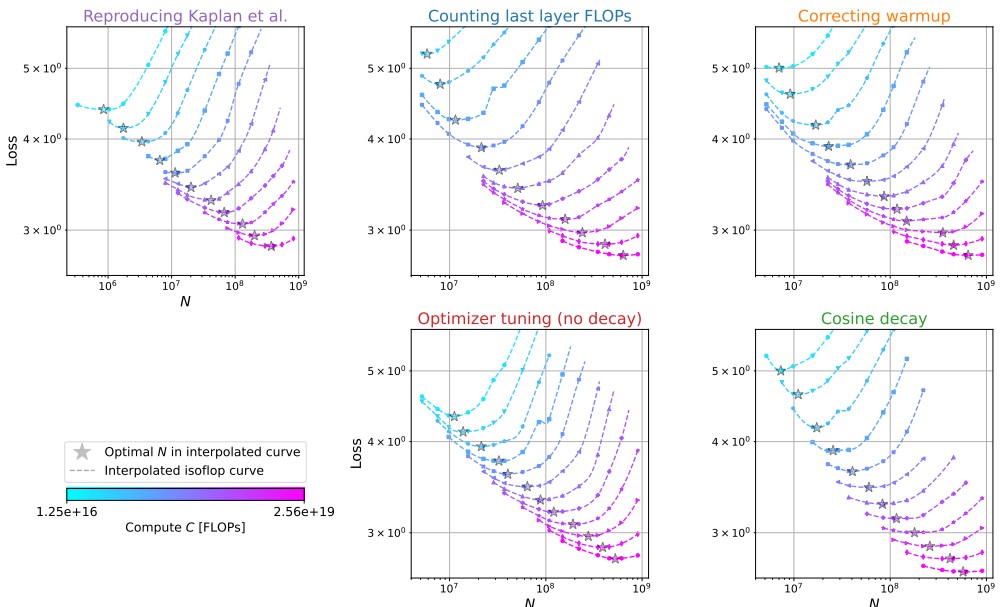

Figure 11: IsoFLOP curves for our main experiments on the OpenWebText2 dataset. The values marked with stars are estimates of $N^\star(C_i)$ for the depicted $C_i$ values. See Appendix D for a related discussion.

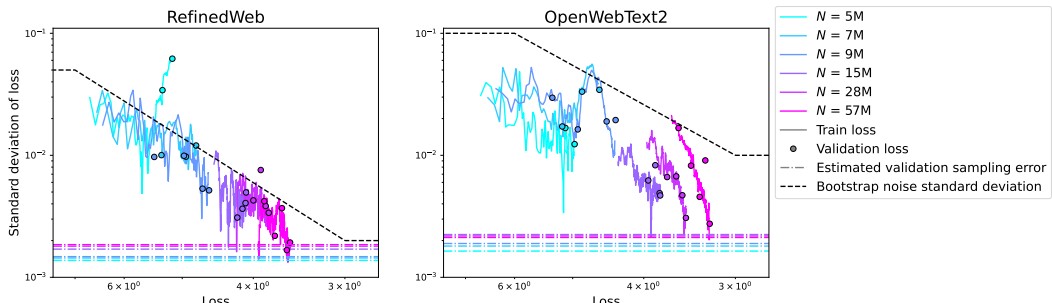

Figure 12: Cross-seed loss standard deviation vs. loss mean, overlaid with our heuristic "bootstrap" noise standard deviation (see Appendix D). The dash-dot lines indicate the standard deviation due to finite validation set size, estimated on the last validation checkpoint and averaged across seeds.

## E  Additional plots for main experiments

In Figure 13 we complement Figure 1 by plotting our observation and power law fits for $D^\star$, $\rho^\star$ and $N^\star$ for all the experiments described in Figure 1. In Figure 14 we reproduce this figure for the OpenWebText2 dataset, showing consistent qualitative and quantitative results.

## F  Ablation of warmup duration and final learning rate value

We preform small scale experiments to verify that our heuristic warmup duration (as discussed in Section 3.3) and the learning rate at the end of cosine decay (as discussed in Section 3.4) have little impact on our results. We train a model with 108M parameters on the RefinedWeb dataset. We use the same hyperparameter as in Table 4, with a training duration of $\sim 2.1$B tokens. We evaluate the models on held-out RefinedWeb data and calculate the validation loss and standard deviation due to sampling as described in Appendix D.

**Warmup duratiom.**   We vary the warmup tokens from $N/4$ to $16N$. We tabulate the results in Table 5, and find that $N$ tokens for warmup is nearly optimal.

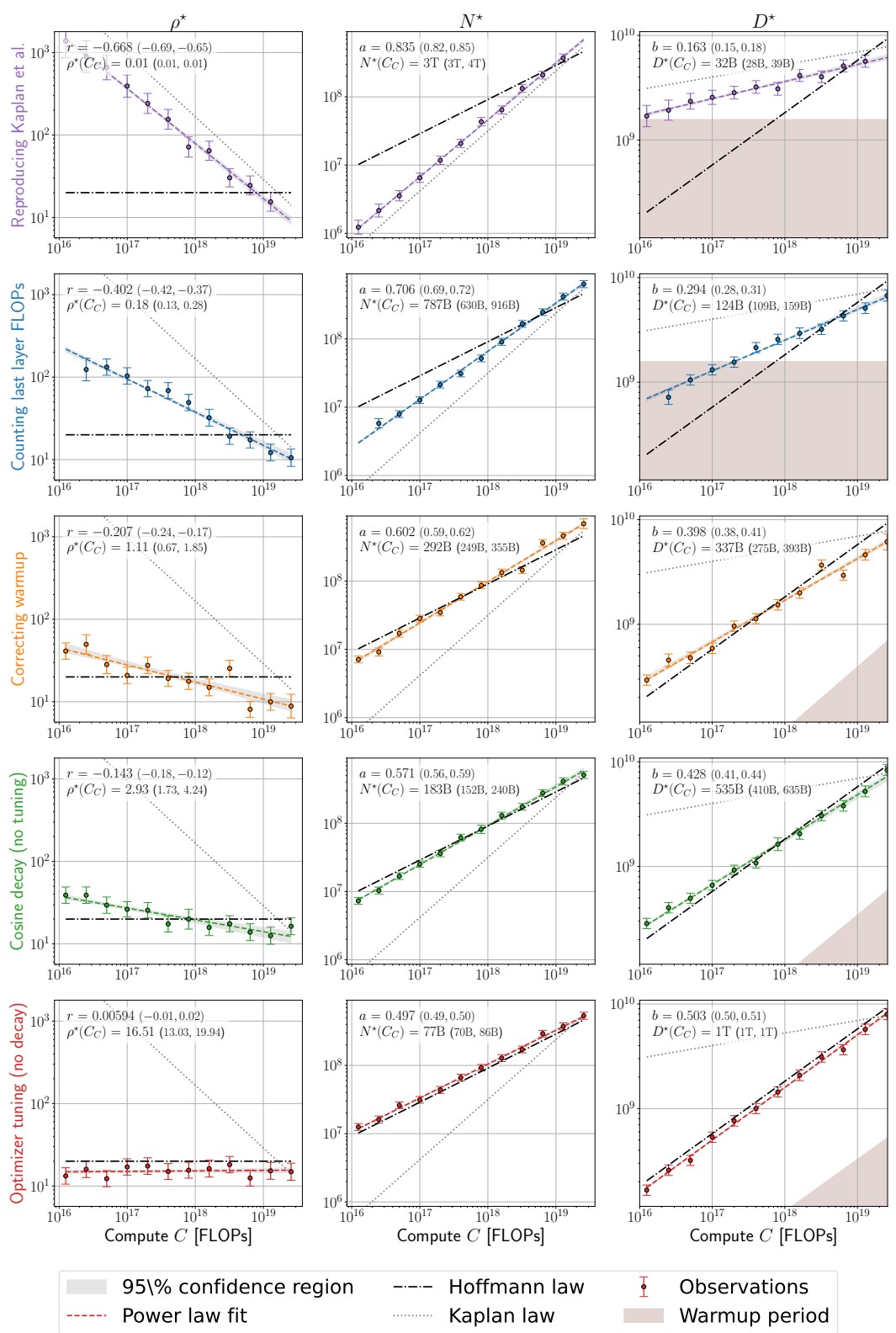

Figure 13: Observations and power law fits of $\rho^\star$, $N^\star$ and $D^\star$ for our main experiments on the RefinedWeb [41] dataset. Here $C_C = 5.88e23$ denotes the compute budget used to train Chinchilla [25].

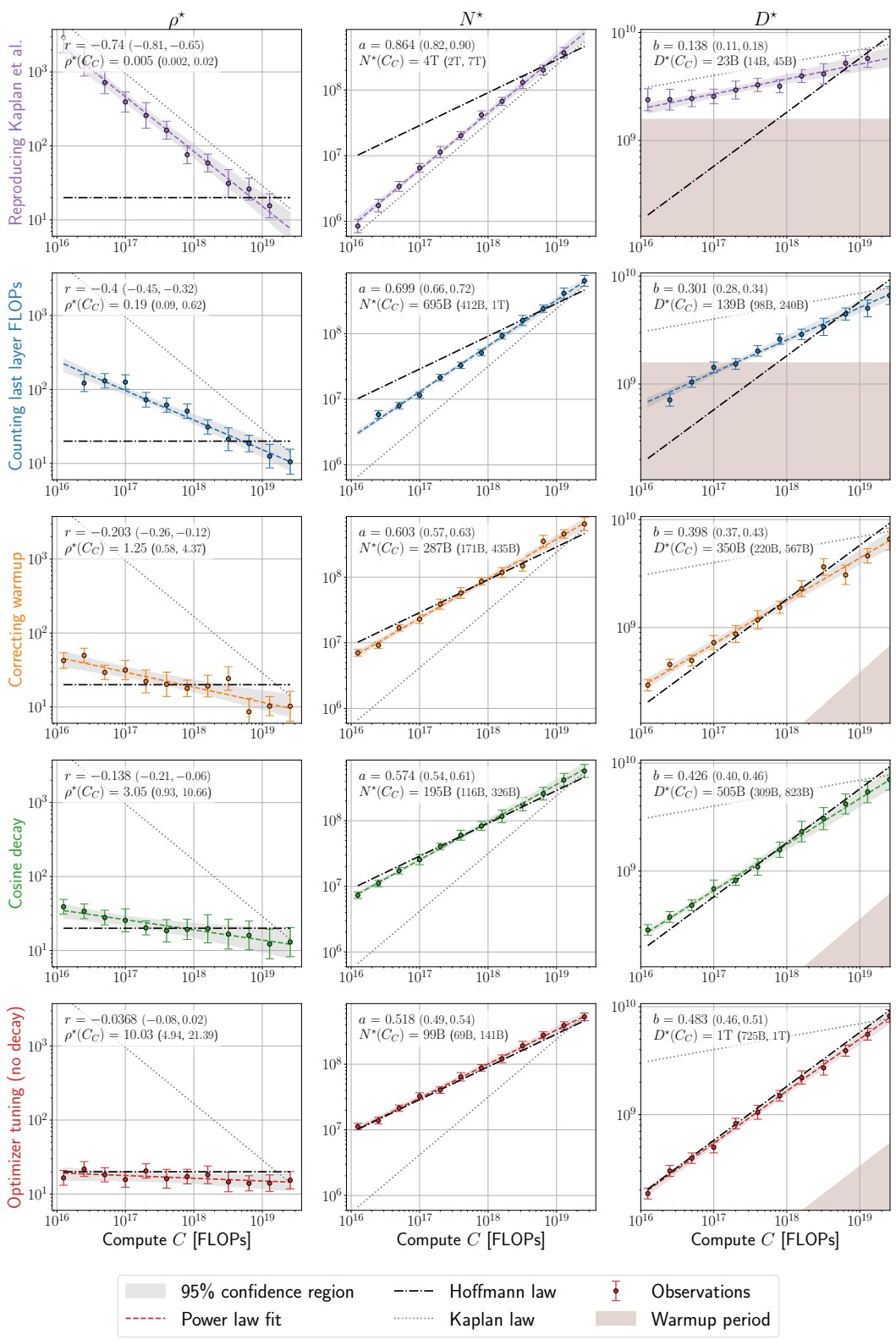

Figure 14: Observations and power law fits of $\rho^\star$, $N^\star$ and $D^\star$ for our main experiments on the OpenWebText2 [18] dataset. Here $C_C = 5.88e23$ denotes the compute budget used to train Chinchilla [25].

Table 5: Ablation of the warmup duration. Durations of up to $4N$ achieve very similar and even slightly better results than the duration $N$ we use throughout the paper. The sampling standard deviation is 0.002 across all experiments.

| Warmup tokens | Loss |
|---|---|
| $N/4$ | 3.546 |
| $N/2$ | 3.542 |
| $N$ (our strategy) | 3.532 |
| $2N$ | 3.528 |
| $4N$ | 3.536 |
| $8N$ | 3.543 |
| $16N$ | 3.586 |

Table 6: Ablation of final learning rate value. The loss is almost not impacted when varying this value from 0.1% to 10% of the peak learning rate. The sampling standard deviation is 0.002 across all experiments.

| Learning rate at the end of the decay | Loss |
|---|---|
| $0.001 \times$ peak learning rate | 3.444 |
| $0.01 \times$ peak learning rate (our strategy) | 3.442 |
| $0.1 \times$ peak learning rate | 3.440 |

**Final learning rate value.** We vary the final learning rate value from $0.1\%$ to $10\%$ of the peak learning rate. Here we train the models with cosine decay and with hyperparameters as described in Table 3. This is the same setting as described in Section 3.4. We tabulate the results in Table 6, and find that the loss is barely impacted by the final learning rate value.

## G  Fitting hyperparmaeters

This section provides a detailed description of our hyperparameter tuning procedure.

### G.1  Full parameter sweep results

We perform an extensive parameter sweep over 6 models from our grid (Table 2) with sizes between 5M and 221M parameters. For each model, we sweep over learning rate and batch sizes, as well as three values of $\beta_2$. We train each model of size $N$ for $20N$ tokens (i.e., following the Hoffmann et al. scaling law) and record the validation loss that the end of training. Overall, our hyperparameter sweep includes 642 training runs, and we perform it on only a single dataset (RefinedWeb). Figure 15 plots all the loss values recorded in the sweep. Compared to analogous plots in Wortsman et al. [58], we observe more sensitivity to the choice of learning rate, particularly for smaller models. We conjecture that this is because all the models in [58] train for the same amount of tokens, so the smaller models become fully convergence for a wide range of learning rates.

### G.2  Estimating the optimal batch size and learning rate via interpolation

To estimate the optimal batch size and learning rate for each model size, we adopt a two-stage interpolation approach. In the first stage, for each model size and batch size, we estimate the optimal learning rate by interpolating (in log-space) the loss as a function of learning rate using Akima [3] interpolation, where for every learning rate we assign the lowest loss obtained from the three values of $\beta_2$. We minimize the interpolant and save its minimizing argument and minimum value. In the second stage, repeat this procedure over the sequence of batch size and interpolated loss pairs, finding an optimal batch size for each model size. To extract an estimate of the optimal learning rate, we simply interpolate the (batch size, minimizing learning rate) sequence and evaluate it at the optimal batch size.

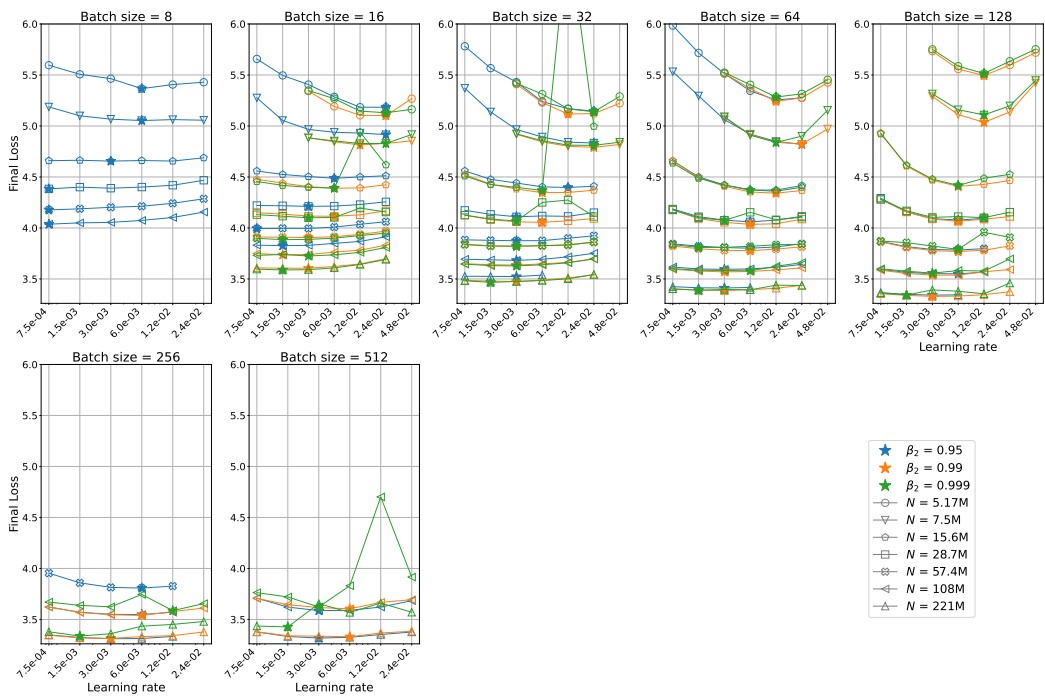

Figure 15: Full hyperparameter sweep results, plotting validation loss after $20N$ training steps as a function of the learning rate for different model sizes $N$, values of $\beta_2$ and batch sizes. Plot design inspired by Wortsman et al. [58].

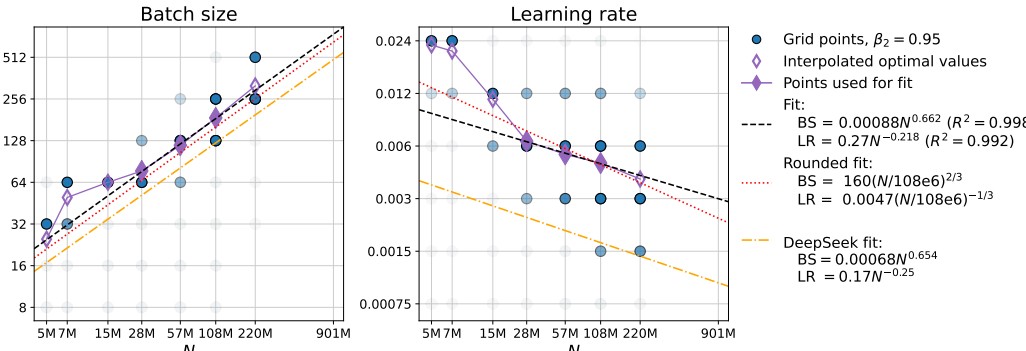

Figure 16: A reproduction of Figure 3 using only $\beta_2 = 0.95$. When we only use this value of $\beta_2$, for small models the optimal batch size saturates, resulting in a less consistent trend. Here the "rounded fit" is the one obtained using all values of $\beta_2$, as in Figure 3.

### G.3 The necessity of tuning $\beta_2$

To demonstrate the importance of tuning $\beta_2$, we repeat the analysis described above except while only considering experiments $\beta_2 = 0.95$. Figure 16 shows the result of this experiment, illustrating that breaks part of the clean scaling trend depicted in Figure 3.

### G.4 Estimating scaling law with ideal tuning

We determine our learning rate and batch size scaling laws by sweeping over hyperparameters for models of sizes $N \leq 108$M with each model trained for $20N$ tokens. As discussed in Section 5.2, this is a limitation as it potentially "bakes in" a preference toward Hoffmann et al. scaling. An ideal tuning strategy would select different hyperparameters for each model size and each compute budget, or equivalently each model size and each token-to-parameter ratio $\rho$.

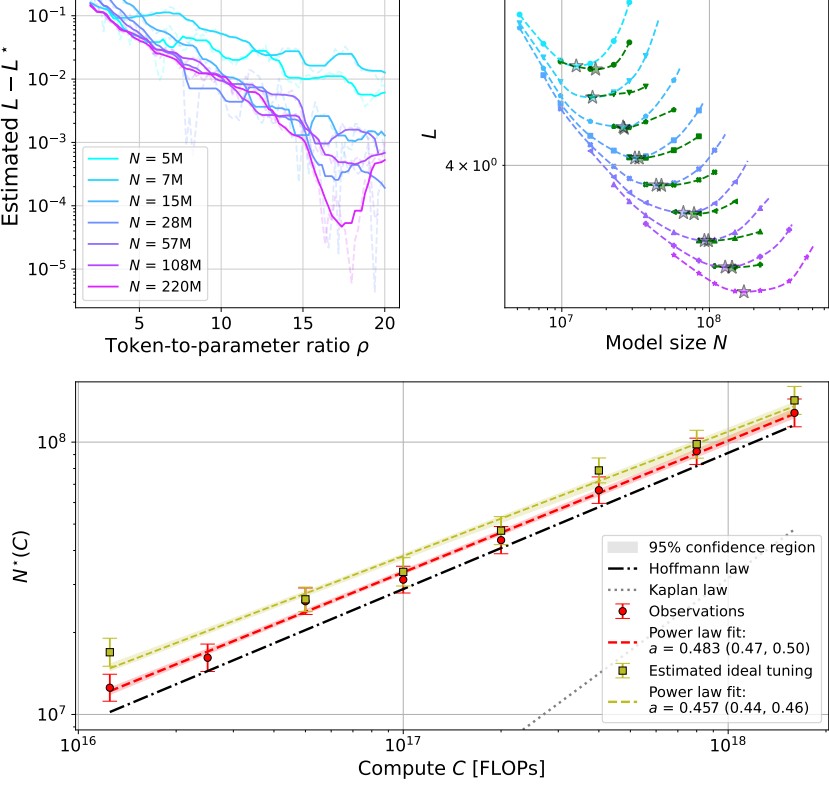

Figure 17: **Top-left:** The estimated excess loss caused by using the hyperparameters in Table 4 instead of the ideal hyperparameters for each model size $N$ and token-to-parameter ratio $\rho$. Light dashed lines show the raw excess loss estimates. To these we apply a median filter of width 2 (in terms of token-top-parameter ratio) and plot the results in solid lines. **Top-right:** The estimated IsoFLOP curves with ideal tuning, obtained by subtracting the excess loss from the actual loss. **Bottom:** Comparing the compute-optimal model sizes obtained from direct observations (with hyperparameters as in Table 4) with our estimate for compute-optimal model sizes given ideal tuning per model.

In this section, we use the training loss data from our hyperparameter sweep to approximate such ideal tuning and estimate its effect on the compute-optimal scaling law. We do so in three steps.

1. **Estimating suboptimality as a function of token-to-parameter ratio.** We estimate the best hyperparameters for $\rho < 20$ using the same interpolation logic as in Section 3.5 but for the training loss after $D = \rho N$ tokens. (We do not consider values of $\rho$ below 2 since they are too close to the warmup period.) Thus, for every value of $N$ and $\rho$, we obtain an estimate of the loss with optimal hyperparameters, denoted $L^\star$. We also use interpolation to estimate the loss under our chosen hyperparameters for each model size (given by Table 4), denoted $L$. Figure 17 top-left shows the (smoothed) estimated suboptimality of our hyperparmaeters, i.e. $L - L^\star$ as a function of $\rho$ for each value of $N$ in the sweep.

2. **Updating IsoFLOP curves.** For all model sizes in Table 4 up to 220M and FLOP values in our grid up to $1.6e18$, we estimate the loss attained by an ideal tuning by subtracting the from our observed loss the smoothed sub-optimality as estimated above at the corresponding value of $\rho$. For model sizes below 220M that are not present in the hyperparameter sweep we interpolate the smoothed sub-optimality based on neighboring model sizes (while keeping the $\rho$ fixed). We include all model size/FLOP combinations with token-to-parameter ratio in between 2 and 30. To estimate the sub-optimality for token multipliers between 20 and 30 (not present in sweep), we extend our smoothed sub-optimality measures symmetrically around $\rho = 20$. Figure 17 top-right

Table 7: Validation loss for the 901M parameter model trained on 14B tokens with different learning rates and batch sizes. The prescribed learning rate and batch size are $0.0024$ and $640$, respectively. The sampling standard deviation is $0.002$ across all experiments.

| LR \ BS | 160 | 320 | 640 | 1280 | 2560 |
|---|---|---|---|---|---|
| 0.0006 | | | 2.962 | | |
| 0.0012 | | 2.970 | 2.947 | 2.954 | |
| 0.0024 | 3.050 | 2.970 | 2.943 | 2.955 | 3.013 |
| 0.0048 | | 2.977 | 2.964 | 2.970 | |
| 0.0096 | | | 2.991 | | |

shows the original IsoFLOP curves (as in Figure 10) along with our estimated loss with ideal tuning.

3. **Re-fitting the scaling law.** To estimate the effect of ideal tuning on our estimate of the compute-optimal exponent $a$, we apply our 'bootstrap' fitting procedure described in Appendix D on the updated IsoFLOP curves described above. To fit the scaling law we only use FLOP values in $\{2^k \cdot 1.25e16\}_{k=0}^7$. For $k > 7$ we do not have data to estimate loss under ideal tuning.

**Conclusions.** The top-left panel of Figure 17 shows that our hyperparameters yield losses within $1e-2$ of the optimal value for models sizes above 15M and $\rho$ values above 10. For smaller models or $\rho$ values, the potential loss reduction from ideal tuning is greater, as is also evident in the top-right panel of the figure. Nevertheless, the top-right panel also shows that the compute-optimal model size (marked by stars on the IsoFLOP curves) do not move much due to the loss reduction. The bottom panel further reveals that for most FLOP values the difference between compute-optimal model sizes is within their estimated standard deviations. It also shows that approximating ideal hyperparameter tuning moves the estimate of the compute-optimal exponent by less than $0.032$, bringing further away from the Kaplan et al. exponent. Furthermore, since learning rate and batch size sensitivities decrease with model size, we expect ideal tuning to have an even smaller effect at larger compute budgets, so we are likely to see even better agreement in $a$ if we introduce larger models to our ideal tuning estimate (which would require extending the hyperparameter sweep to larger models as well). Indeed, dropping the two smallest FLOP values from the power law fit (that is, using FLOP values $\{2^k \cdot 1.25e16\}_{k=2}^7$) yields exponent $a = 0.489$ for the original observations and exponent $a = 0.5$ for ideal tuning. Overall, we estimate that ideal hyperparameter tuning would produce similar results to our scaling-law-based hyperparameter choices.

### G.5 Validating the hyperparameter scaling laws on a larger compute scale

We extend the training of our largest model (with 901M parameters) to $\sim$14B RefinedWeb tokens (the compute-optimal value for that model size according to our scaling law), and test whether our predicted batch size and learning rate are optimal at this scale. In particular, we compare the learning and batch size prescribed in Table 4 (0.0024 and 640, respectively) to 12 configurations. In 8 of them we very either the learning rate or the batch size, and in 4 of them we very both. We evaluate the models on held-out RefinedWeb data and calculate the validation loss and standard deviation due to sampling as described in Appendix D. The results are shown in Table 7. We find that the prescribed learning rate and batch size are indeed optimal, as they yield the lowest validation loss.

## H Reproducing the adjusted Kaplan et al. scaling law

Figure 18 shows that by reintroducing the FLOP count and long warmup issues from Sections 3.1 to 3.3 but training with optimized hyperparameters, we recover the "adjusted" form of the Kaplan et al. [30] compute-optimal scaling law, given by $1.3e9 \cdot (C/8.64e19)^{0.73}$. Kaplan et al. [30] derive this scaling law by theoretically compensating for the fact that they used a too-large batch size for smaller models. It is reassuring to observe that when we add back the other issues we have identified but appropriately decrease the batch size via parameter tuning, we obtain very close agreement with this adjusted scaling law.

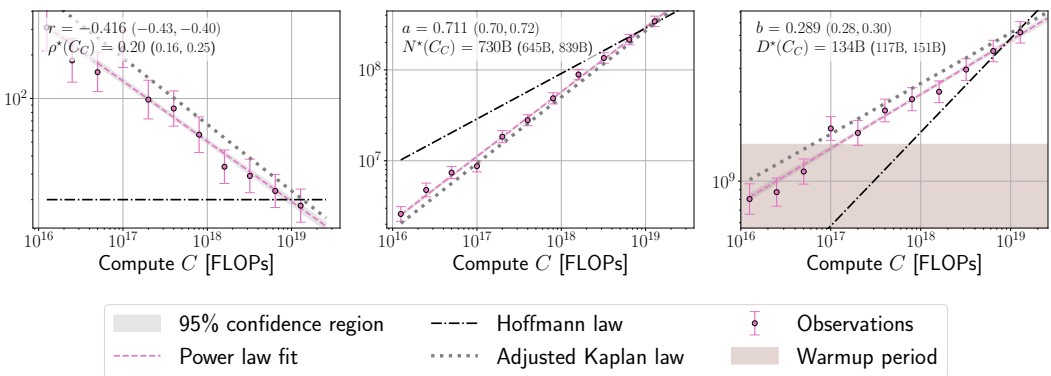

Figure 18: The optimal model size $N^\star$ as a function of the compute budget $C$, for models trained with the tuned hyperparameters of Section 3.5 but with the long warmup and discounting the model head in the FLOP counts as done in Kaplan et al. [30].

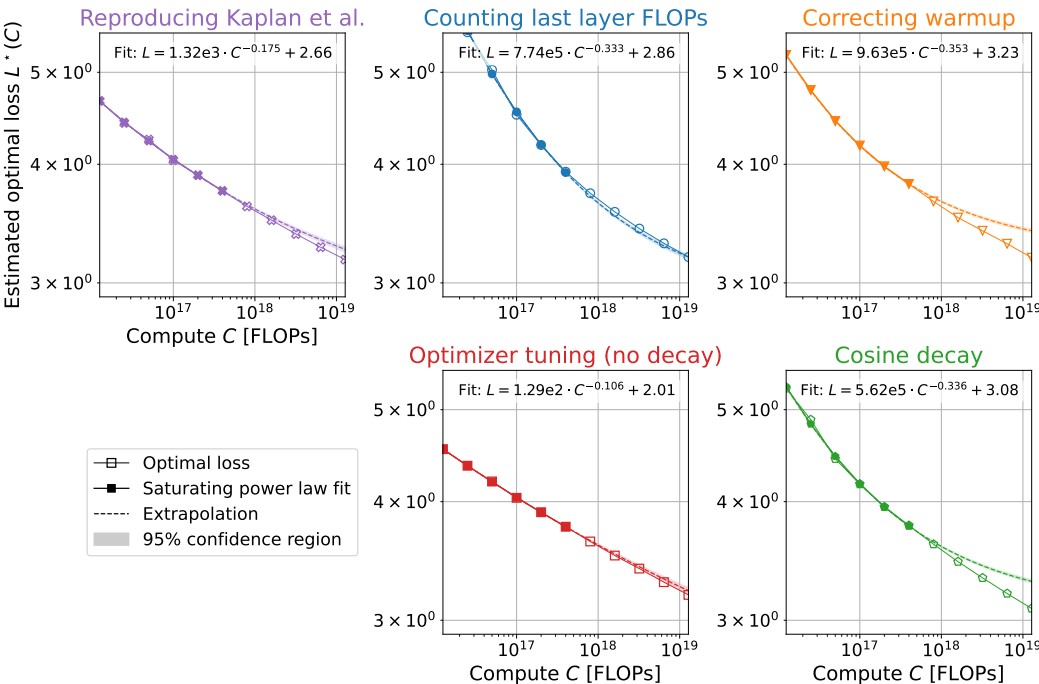

Figure 19: An expanded version of Figure 4 showing a saturating power law fit for each experiment.

# I   The compute-optimal loss

**Extending Figure 4.**   In Figure 19 we fit a saturating power law of the form $L(C) = E + L_0 C^{-\ell}$ to each of the compute-optimal loss curves in our experiments. As the figure shows, the fit is predictive only for the experiment where hyperparameters are tuned. We fit the saturating power law similarly to Hoffmann et al. [25], by minimizing the Huber prediction loss for $\log L(C)$ over $\ell, \log(E)$ and $\log(L_0)$.

**Results on the OpenWebText2 dataset.**   We perform the same analysis on the OpenWebText2 dataset. In Figure 20 we show the fit of the saturating power law to the compute-optimal loss curves for the OpenWebText2 dataset, and in Figure 21 we show the OpenWebText2 version of Figure 5. The loss scaling law is predictive for the OpenWebText2 dataset as well.

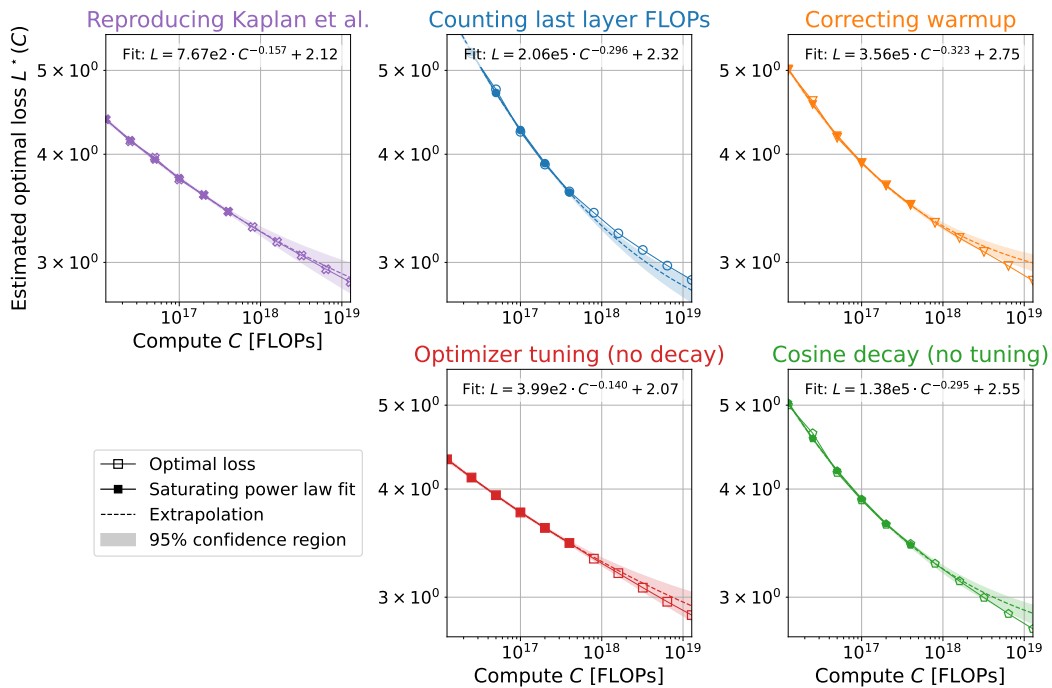

Figure 20: Reproduction of Figure 19 on the OpenWebText2 dataset.

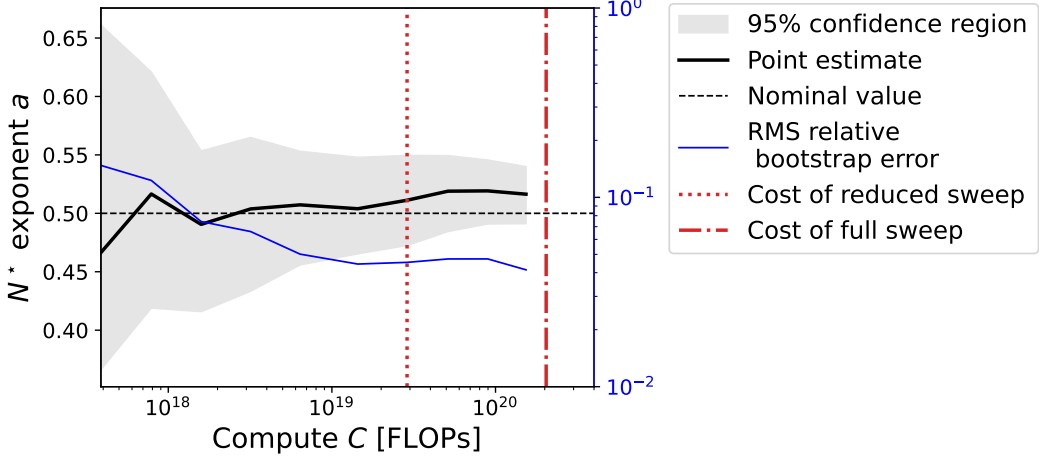

Figure 21: Reproduction of Figure 5 on the OpenWebText2 dataset.

## J The computational cost of our experiments

We now discuss the calculation of the cost of each of our main experiments, comparing fixed learning rate schedules (constant or fixed-length cosine as in Kaplan et al. [30]) with a cosine schedule tailored for each. Each of our experiments consists of directly estimating $N^\star(C)$ for a grid of $C$ values of the form $C_k = 2^k \cdot 1.25e16$ FLOPs and $k$ going from 0 to $K = 11$. With a cosine learning rate schedule, each value of $C$ requires distinct training runs, so the cost of the experiment is $\sum_{k=0}^{K} m_k C_k$, where $m_k$ is the number of models we train for $C_k$ FLOPs—between 6 and 7 in our experiments. A constant learning rate schedule offers savings since we can extract performance at different FLOP values from the same run, so the cost of the experiment is $\sum_{k=0}^{K} m'_k C_k$ where $m'_k$ is the number of models we train for *at most* $C_k$ FLOPs. At the maximum budget we have $m'_K = m_K$ between 6 and 7, but for all smaller $k < K$ we have $m'_k$ between 0 and 2 (typically 1). Thus, we save $\sum_{k=0}^{K-1} (m_k - m'_k) C_k \approx \sum_{k=0}^{K-1} m_k C_k$, which for our doubling grid of $C$ is roughly half the cost. For a fair comparison, when empirically summing over the cost of our experiments omit runs where the number of tokens is more than 100 times the model size or where loss is more than 1 nat above the optimal loss for the compute budget since they do not contribute to the analysis and when experimenting with a cosine schedule we were more careful not execute them.

