# OpenReview forum: "Resolving Discrepancies in Compute-Optimal Scaling of Language Models"
_NeurIPS.cc/2024/Conference — NeurIPS 2024 spotlight_

### Official Review · Reviewer_CKuz · 2024-07-06

**Soundness:** 3
**Presentation:** 3
**Contribution:** 2
**Rating:** 5
**Confidence:** 4

**Summary:**

This paper aims to resolve the discrepant conclusion about compute-optimal model size drawn from (Kaplan et al., 2020) and (Hoffmann et al., 2022), two famous scaling-law papers that have guided the exploration of early large language models. The authors concluded that the discrepancies arise from last-layer FLOPS and warmup steps. And contrary to the claims of (Hoffmann et al., 2022), learning rate decay is not essential. As a byproduct, the authors also develop strategies to select hyperparameters, including optimal learning rates and batch sizes, and emphasize the need to use larger beta_2 when training with small batch sizes using Adam.

**Strengths:**

1. This paper targets an interesting and important topic on the discrepancy of scaling laws. Given that scaling laws guide the design decision to train modern large language models, solid results on this topic can confirm appropriate scaling strategies and help consolidate the reliability of scaling laws for applications such as large-scale performance prediction.
2. This paper presents extensive and detailed experiments to show how to align two scaling laws for compute-optimal model sizes.
2. The discussion on hyperparameters selection, including learning rates, batch sizes, warmup durations, and beta_2 of Adam provide helpful guidance on reproducing scaling law results.

**Weaknesses:**

1. Discrepancies in scaling laws indicate concerns about the correctness of scaling laws. Most of the paper discusses how to resolve the discrepancies between two scaling laws but has not fully addressed the concerns about the correctness of the two scaling laws as only limited results (only Fig. 4 with one dataset) show resolving the discrepancies generally leads to accurate prediction in large-scale experiments.
2. The conclusion of this paper is not clear enough. It would be clearer if the authors could conclude a standard pipeline to reproduce the scaling law experiments and enable reliable predictions with this pipeline.
3. This paper focuses on L(C), the compute optimal scaling laws, instead of L(N, D), which is a more general scaling laws that consider both model sizes and training data. Given that the latest large language models are overtrained to reduce inference costs, limiting to the compute-optimal setting may be less impactful.

**Questions:**

1. One of my concerns on scaling laws is that the need for careful hyperparameter selection may impede its applications. Firstly, it makes fitting a law extremely complex, and making mistakes that lead to unreliable predictions is easy. Also, the need to tune hyperparameters increases the costs to fit a law, indicating that we need accurate extrapolation to orders of larger magnitude to pay off the costs. I would like to learn about the authors' comments on potential strategies to stabilize the process and reduce the costs of hyperparameter tuning.

2. Given that many scaling laws exist, and they are discrepant possibly because of different experimental settings or assumptions, could you summarize the condition for the scaling laws in this paper?

**Limitations:**

This paper only explores the compute-optimal settings and provides limited extrapolation results based on their findings (only one dataset and only scales up to 3e19 flops).

---

> ### Author Rebuttal · Authors · 2024-08-06
>
> We thank the reviewer for the detailed review, finding that our paper presents extensive and detailed experiments and provides helpful guidance on reproducing scaling law results. We address the weaknesses and questions below - the main issue appears to be a lack of clarity regarding which scaling law (Kaplan or Hoffmann) is “correct.” We hope our answers clarify this, and we look forward to engaging in constructive discussion about how to introduce these clarifications to the revision best.
>
> **Unclear which scaling law is correct**
>  Conclusively deciding the correctness of compute-optimal scaling laws is a difficult problem due to computational barriers. Nevertheless, our paper adds to a growing body of evidence that Hoffmann et al. (“Chinchilla”) scaling law is more accurate than the Kaplan et al. scaling. Let us explain this in detail:
>
> * The difficulty of validating scaling laws. Directly testing the prediction of a compute-optimal scaling law at the largest compute budget in which one can afford to train is almost impossible by definition, as such verification would require multiple largest-scale training runs. Indeed, neither Kaplan et al. nor Hoffmann et al. directly verify the compute-optimality of extrapolations of their scaling laws.
>
> * Theoretical evidence in favor of the Hoffmann et al. scaling law. The compute-optimal exponent $\alpha=0.5$ found by Hoffmann et al. has the elegant interpretation that model size and data size should scale with constant proportion. Recent theoretical work has proven the same scaling holds for simple, analytically tractable settings under fairly broad assumptions. The paper “4+3 Phases of Compute-Optimal Neural Scaling Laws” by Paquette et al. shows this for a random-feature linear regression setting, while the paper “Information-Theoretic Foundations for Neural Scaling Laws” by Jeon and Van Roy shows this for data generated by infinite-width 2 layer ReLU networks using information-theoretic arguments. These works hint at (and, in the former case, explicitly conjecture) the possibility of the optimality of proportional scaling being a broader, universal phenomenon.
>
> * Prior empirical evidence in favor of the Hoffmann et al. scaling law. Hoffmann et al. nevertheless provide strong evidence in favor of their scaling law - the Chinchilla 70B model outperformed the larger Gopher model that was trained using similar compute, architecture, and data, as well as other other larger contemporary models. As we discuss in our related work section, in language modeling, subsequent work [12, 22] has produced scaling laws more similar to Hoffmann et al., and other papers rely on the validity of this scaling as a starting point [14, 29]. Therefore, we believe that - for the specific context of next token prediction with Transformer decoder-only models - it is established that the Hoffmann et al. scaling law is the more accurate one; what was missing prior to our work is an explanation for why this is so.
>
> * New evidence in favor of the Hoffmann et al. scaling law from our work. Our paper attributes the difference between the Kaplan and Hoffman scaling laws to what are arguably methodological errors in Kaplan et al.: Incorrect FLOP counting, overly-long warmup period, and suboptimal optimizer configuration. Furthermore, we show that the Hoffmann scaling law extends to non-decayed learning rate schedules. In addition to providing useful lessons for future scaling law studies, our findings further affirm that the Hoffmann et al. scaling is the “more correct” one.
>
> **Unclear conclusion from the paper**
> We hope the discussion above helps position our paper in context and thus clarify its conclusion. Moreover, our final reproduction of the Hoffmann scaling constitutes a well-defined pipeline that enables reliable predictions - the training code is based on the open-source open_lm library, and all of the configurations and analysis code are included in our submission.
>
> **Limited impact of compute-optimal setting**
> We agree that studying over-trained language and more broadly the individual effect of scaling $N$ and $D$ are important topics that warrant further research. However, note that the very concept of overtraining is defined relative to compute-optimal training. Thus, a better understanding of the compute-optimal setting is an essential stepping stone for broader studies as well. In particular, our conclusions about FLOP counts, warmup, and hyperparameter tuning should be relevant beyond the strictly compute-constrained setting.
>
> **Limited study of predictive accuracy**
> Regarding the concern that we study predictive accuracy only in one figure (Figure 4) and one dataset, note that Figure 5 also studies the predictive accuracy as we vary the amount of compute used to fit the scaling laws. In the attached pdf we further study predictive accuracy in two ways:
>
> 1. We reproduce Figures 4 and 5 for the OpenWebText2 dataset (this involves simply running the analysis code attached to our submission with different parameters).
>
> 2. Following suggestions from Reviewers zsos and CKuz, we trained a model for ~8e19 FLOPs (4x our previous largest budget) under our predicted compute-optimal settings and hyperparameters. We show that the resulting loss agrees fairly well with our extrapolation in Figure 4, with further improvements as we fit it on slightly larger budgets of up to 1.6e18 FLOPs. Moreover, we confirm the optimality of our predicted learning rate and batch size - see the response to reviewer CKuz for details.
>
> Thus, even though the focus of our paper is explaining the scaling law discrepancy and not measuring predictive accuracy, we study the latter to a greater extent than the influential papers of Kaplan et al. and Hoffmann et al. We nevertheless agree that the two extensions described above will strengthen our paper and will include them in the revision.
>
> (Due to length constraints we answer the two questions in the review in the comment below.)

---

> ### Author Response · Authors · 2024-08-07
>
> **Question 1: hyperparameter sensitivity**
> As we remark in the limitations section, we believe that scaling up the models reduces optimization hyperparameter sensitivity, which would explain why Hoffmann et al. were able to obtain their scaling law without very careful parameter tuning. Moreover, Figure 10 in our paper shows that scaling up the model size leads to higher robustness to learning rate and batch size.
>
> Finding ways to make smaller-scale models also require less per-scale tuning is an important problem - maximal update parameterization (𝜇P) is a promising direction still under active research; we will discuss it in the revision.
>
> **Question 2: Condition for scaling laws**
> We focus on decoder-only Transformer models, trained for next token prediction with the log loss. This focus is the same as in Kaplan et al. and Hoffmann et al., the two papers that presented the scaling laws in question. Beyond that, we argue that the discrepancy between the two scaling laws is not due to a difference in assumptions, but rather due to differences that are largely methodological errors in Kaplan et al.

---

> ### Author Response · Authors · 2024-08-12
>
> Dear reviewer,
>
> Before the author-reviewer discussion period ends, we would like to know if our responses above fully address your questions and concerns. If so, we kindly ask you to reconsider our paper’s score.

---

### Official Review · Reviewer_Kjsg · 2024-07-09

**Soundness:** 4
**Presentation:** 4
**Contribution:** 4
**Rating:** 7
**Confidence:** 4

**Summary:**

This paper studies the discrepancy between the scaling laws for Kaplan et al. and Hoffmann et al. and identifies three factors behind it: last taking last layer cost into account, warmup duration, and tuning optimizer hyperparams with model scale. They show the transition from Kaplan et al. scaling laws before correcting for these effect to Hoffmann et al. scaling laws after correcting for these effects.

**Strengths:**

The paper studies the important question of scaling laws for language models thoroughly (with the limitation of model scale). This fills an important missing piece in the literature and places the scaling law research on a firmer grounding. While running these experiments they also discover additional interesting observations such as the importance of tuning beta2 at smaller batch sizes.

**Weaknesses:**

The main limitation of this work as also discussed by the authors is the scale of the models use (< 1B).

A more stronger instance of this limitation is that the paper uses experiments on model of size 5M to 100M to fit scaling laws for learning rate and batch size, it is unclear if these laws would hold up for large models. It would have been useful if the paper did an check of this optimality at a significantly larger scale (~1B, not just 200M). This is less of a concern for the runs in the paper since all of them have model sizes <=1B and since Adam is known to be stable (Wortsman et al. 2023) across a range of learning rates.

**Questions:**

1. Is the sweep described in appendix E done with constant learning rate or cosine decay?
2. While the papers fixing of warmup to be always less than the model size is certainly a step in the right direction as compared to Kaplan et al. is there. a reason to believe that this is optimal? or that the scaling law would not change with change with the warmup duration?

**Limitations:**

Yes.

---

> ### Author Rebuttal · Authors · 2024-08-06
>
> We thank the reviewer for the useful questions and kind words. We were glad to read that you find our paper studies the important question of scaling laws thoroughly and fills an important missing piece in the literature. Below, we address the weaknesses and questions raised in the review.
>
> **Scale of experiments**
> Following the reviewer’s suggestion, we extend the training of our largest model (with 901M parameters) to ~14B RefinedWeb tokens (the compute-optimal value for that model size according to our scaling law), and test whether our predicted batch size and learning rate are optimal at this scale. In particular, we compare the learning and batch size prescribed in Table 4 (0.0024 and 640, respectively) to 8 configurations, each varying either the learning rate or the batch size. We evaluate the models on held-out RefinedWeb data and calculate the validation loss and standard deviation due to sampling as described in L583. We tabulate the results below:
>
> |   Learning Rate | Validation Loss   |
> |----------------:|:------------------|
> |          0.0006 | 2.962 ± 0.002     |
> |          0.0012 | 2.947 ± 0.002     |
> |          0.0024 | 2.943 ± 0.002     |
> |          0.0048 | 2.964 ± 0.002     |
> |          0.0096 | 2.991 ± 0.002     |
>
> |   Batch Size | Validation Loss   |
> |-------------:|:------------------|
> |            160 | N/A    |
> |           320 | N/A     |
> |           640 | 2.943 ± 0.002     |
> |           1280 | 2.955 ± 0.002     |
> |           2560 | 3.013 ± 0.002     |
>
> These results indicate that we accurately predict the optimal hyperparameters at the ~1B scale. The runs with batch sizes 160 and 320 did not complete by the rebuttal deadline, and we will attach them as they arrive; intermediate loss evaluations suggest they will not outperform our predicted optimal batch size.
>
> **Question 1: learning rate scheduler in sweep**
> We use a constant learning rate in our sweep experiments, with a warmup duration of N (number of parameters) tokens.
>
> **Question 2: warmup heuristic**
> To address the reviewer’s question, we perform a small-scale experiment where we train a $N$=108M parameter model on the RefinedWeb dataset for $20N=$~2.1B tokens, using the hyperparameters in Table 4 (for the appropriate model size) but varying the warmup duration from $N/4$ to $16N$. We evaluate the models on held-out RefinedWeb data and calculate the validation loss and standard deviation due to sampling as described in L583. We tabulate the results below:
>
> | Warmup tokens   | Validation loss   |
> |:---------|:------------------|
> |$N/4$    | 3.546 ± 0.002     |
> | $N/2$     | 3.542 ± 0.002     |
> | $N$        | 3.532 ± 0.002     |
> | $2N$       | 3.528 ± 0.002     |
> | $4N$       | 3.536 ± 0.002     |
> | $8N$       | 3.543 ± 0.002     |
> | $16N$      | 3.586 ± 0.002     |
>
> The results show that a warmup duration of $N$ is nearly optimal, with durations of up to $4N$ achieving very similar and even slightly better results. We conclude that the training loss is fairly insensitive to the precise warmup duration as long as it stays in a reasonable range. Consequently, we believe that our scaling laws are not strongly dependent on the particular choice of warmup duration.

---

> > ### Author Response · Authors · 2024-08-08
> > **Complete results of batch size sweep**
> >
> > We tabulated the full results of the batch size sweep below:
> > |   Batch Size | Validation Loss   |
> > |-------------:|:------------------|
> > |            160 | 3.050 ± 0.002      |
> > |           320 | 2.970 ± 0.002      |
> > |           640 | 2.943 ± 0.002     |
> > |           1280 | 2.955 ± 0.002     |
> > |           2560 | 3.013 ± 0.002     |
> >
> > The results show that our predicted batch size is optimal among the values we have tested.

---

> > > ### Comment · Reviewer_Kjsg · 2024-08-08
> > > **Response to the authors.**
> > >
> > > Would it be possible to show the performance on the grid [320, 640, 1280] x [0.0012, 0.0024, 0.0048]. This is important since batch size and learning rates have strong interactions. Single runs should be sufficient since the standard deviation is low.

---

> > > > ### Author Response · Authors · 2024-08-11
> > > >
> > > > We tabulated the results of our experiments of varying both learning rate and batch size below:
> > > > |   LR \ BS | 320           | 640           | 1280          |
> > > > |----------:|:--------------|:--------------|:--------------|
> > > > |    0.0012 | 2.970 ± 0.002  | 2.947 ± 0.002 | 2.954 ± 0.002 |
> > > > |    0.0024 | 2.970 ± 0.002  | 2.943 ± 0.002 | 2.955 ± 0.002 |
> > > > |    0.0048 | 2.977 ± 0.002 | 2.964 ± 0.002 | 2.970 ± 0.002  |
> > > >
> > > > Indeed, our predicted hyperparameters are optimal.

---

> > > > ### Author Response · Authors · 2024-08-12
> > > >
> > > > Since our 1B model experiments address the main limitation pointed out in your review, would you consider raising our paper’s score?

---

> ### Author Response · Authors · 2024-08-08
>
> We have queued the 4 additional runs requested and hope they will complete before the discussion period is over. However, we note that Figure 10 suggests that, for larger models, the interaction between batch size and learning rate is not very strong.

---

### Official Review · Reviewer_comx · 2024-07-14

**Soundness:** 2
**Presentation:** 4
**Contribution:** 3
**Rating:** 6
**Confidence:** 4

**Summary:**

The paper provides an explanation for the discrepancies between the scaling laws of Kaplan et al. and Hoffman et al. The authors start by reproducing the scaling laws of Kaplan at al. They then introduce incremental changes in the methodology: accounting for the last layer computational cost, setting a more reasonable learning rate warmup, and more thorough scale-dependent hyperparameter optimization. After introducing these changes, they recover the scaling laws of Hoffman et al. The paper provides evidence that careful learning rate decay is not necessary to resolve the differences between these two scaling laws. The paper demonstrates the importance of such scale-dependent hyperparameters for obtaining reliable scaling laws with small computational budgets.

**Strengths:**

I believe the paper to be novel — I am unaware of prior work aiming to explain the differences between the scaling laws of Kaplan et al. and Hoffman et al., beyond the hypothesis put forth by Hoffman et al. which this paper directly addresses.

The paper is very clearly written. Its experimental set-up is well-motivated in relation to the prior work of Kaplan et al. and Hoffman et al. The experiments are thorough, and the methodology sound.

Beyond the scientific interest of studying why Kaplan et al. and Hoffman et al. arrived at such different scaling law coefficients, this paper provides valuable insights for researchers studying the scaling of language models, particularly regarding the importance of scale-dependent hyper-parameter optimization for small computational regimes.

**Weaknesses:**

Hoffman et al. suggests that their fitted scaling law parameters differ from those of Kaplan et al. because Kaplan et al. include very sub-optimally trained models in their analysis, which bias the fitted parameters. That is, for certain (N, D, L) tuples, the loss L is substantially overestimated. They suggest that this is mainly due to the poor learning rate schedule used by Kaplan et al. Contrary to what the paper under review claims at times, I view this paper as supporting the broad hypothesis of Hoffman et al. rather than disproving it. Starting with training choices similar to those of Kaplan et al., the authors are able to reproduce their scaling law. They then show that improving these training choices — via a more reasonable warmup schedule, batch size, learning etc. — yields a scaling law that more closely resembles that of Hoffman et al. That is, the poorer training hyper parameters of Kaplan et al. explain the difference with Hoffman et al. While Hoffman et al. emphasize the learning rate schedule, I don’t read their paper as claiming that learning rate decay is necessary to obtain their scaling law —as the authors claim at times—, but rather that the poor learning rate scheduler used by Kaplan et al. is sufficient to mess up their derived scaling law. To me, the hypothesis of Hoffman et al. is consistent with the existence of different hyper parameter choices for which similar (N, D, L) is attainable with a constant learning rate — they do not claim that is is not possible.

The authors decouple “warmup” from “decay”, whereas to me the learning rate schedule comprises both. Thus, when Hoffman et al. say that they set “the learning rate schedule to approximately match the number of training tokens”, I imagine that they mean adapting both the warmup and the decay period (e.g., to use heuristics such as warmup for 5% of the total training steps). Unfortunately, Hoffman et al. do not give details on what warmup schedule they used. However, looking at Figure A1, it is clear that the warm-up period that they use is a small fraction of the total number of training steps — unlike Kaplan et al. Broadly, I find it misleading to say “Counter to a hypothesis of Hoffmann et al., we find that careful learning rate decay is not essential” when Hoffman et al. say “Our work differs from Kaplan et al. (2020) in several important ways. First, the authors use a fixed number of training tokens and learning rate schedule for all models. […] In contrast, we find that setting the learning rate schedule to approximately match the number of training tokens results n the best final loss regardless of model size” — notice the difference in language, “learning rate schedule” (which is broadly understood to include also warmup) instead of “learning rate decay”.

I ask the authors to please better scope their claim: “Counter to a hypothesis of Hoffmann et al. [21], we find that careful learning rate decay is not essential for the validity of their scaling law” — Hoffman et al. do not claim necessity, and they refer to “learning rate schedule” rather than only learning rate decay.

Lastly, the authors may be underestimating the value of learning rate decay, since they only consider models that are reasonably close to Chinchilla-optimal. When considering, for the same model size, very different number of training tokens, cosine decay may play a more important role, at least if all other hyper parameters are kept fixed. The authors are aware of this and discuss it in the limitations section.

**Questions:**

For Section 3.5, do you warmup the learning rate? How so?

Hoffman et al. decay learning rate to 10% of peak, why do you use 1% of peak?

**Limitations:**

The authors adequately discuss the limitations of their work.

---

> ### Author Rebuttal · Authors · 2024-08-06
>
> We thank the reviewer for the detailed comments and for finding our paper novel, clearly written, and our experiments well-motivated, sound,  thorough, and providing valuable insights. Below we address the main concern regarding the conjecture in Hoffmann et al., as well as the two additional questions. With these points addressed we hope the reviewer will consider increasing our paper’s score.
>
> **The conjecture in Hoffmann et al.:**
> We agree that the term “learning rate schedule” comprises both the warmup and decay components, and that it is plausible to interpret the “Modeling the scaling behavior” paragraph in Hoffmann et al. §2 as conjecturing that both components contribute to the scaling law discrepancy. However, based on Appendix A of Hoffmann et al. as well as the subsequent literature discussing this paper, we believe that Hoffmann et al. emphasize the decay component of the learning rate schedule in their conjecture.
>
> _A closer look at Hoffmann et al. Appendix A:_ In §2, Hoffmann et al. refer to Figure A1 to provide evidence for the claim that “setting the learning rate schedule to approximately match the number of training tokens results in the best final loss regardless of model size.” However, in Figure A1 they only vary the decay component of the learning rate, keeping the warmup. Hence, it appears that — in their conjecture regarding the scaling law discrepancy — Hoffmann et al. equate learning rate schedule with learning rate decay.
>
> _Subsequent literature:_ The following sources treat the conjecture of Hoffmann et al. as pertaining to only learning rate decay, suggesting that the above interpretation of Hoffmann et al.’s conjecture is common in the literature:
>
> * Hu et al. [44, §4.1] write that
>   > … Hoffmann et al. (2022) make a key observation that setting $T > S$ results in dropped performance while setting $S = T$ results in improved training efficiency, confirming that the learning rate shouldn’t be kept high throughout the training.
>
>   They do not mention aspects of the learning rate schedule other than the decay period.
>
> * In “Scaling laws in compute-optimal training” (to be cited in the revision) Hägele et al. write that
>   > Importantly, the Chinchilla project (Hoffmann et al., 2022) showed that the cosine schedule achieves optimal loss only when the cycle length matches the training duration, but underestimates the model performance during training.
>
>   The cosine cycle length controls the learning rate decay, not the warmup.
>
> * A LessWrong blog post titled “New Scaling Laws for Large Language Models” discusses the scaling law discrepancy and states that
>
>   > It looks like OpenAI used a single total annealing schedule for all of their runs, even those of different lengths. This shifted the apparent best-possible performance downwards for the networks on a non-ideal annealing schedule. And this lead to a distorted notion of what laws should be.
>
>   Again, no mention of warmup.
>
> We nevertheless agree that the interpretation of Hoffmann et al.’s conjecture is subtle and requires further justification and clarification, which we are happy to add to the revision. In particular, in line 8 (the abstract) we will write “Counter to a hypothesis implied in Hoffmann et al.” and in line 30 we will remove the word “decay” from “tailoring the learning rate decay schedule”. In addition, we will add an appendix with a detailed discussion about the conjecture of Hoffmann et al., along the lines of our response above.
>
> **Question 1: Warmup in hyperparameter sweep**
> We use the same heuristic for choosing warmup duration as discussed in section 3.3, equating the warmup duration (in tokens) with the number of model parameters. We will add this detail to Appendix E.
>
> **Question 2: Cooldown value**
> We decayed the learning to 0.01 of its peak value following Gadre et al. [14], which was the source of most of the initial hyperparameters used in our experiments. Following the reviewer’s question, we ran a small-scale experiment to test the potential effect of using a different final learning rate. In particular, we train a model with 108M parameters on the RefinedWeb dataset. We use the same hyperparameter as in Table 3, with a training duration of ~2.1B tokens. We use three different values for the final learning rate at the end of the cosine decay - 0.1%, 1%, and 10% of the peak learning rate. Finally, we evaluate the models on held-out RefinedWeb data and calculate the validation loss and standard deviation due to sampling as described in L583. We tabulate the results below:
>
> |   Learning rate at the end of the run | Validation loss   |
> |-----------:|:------------------|
> |     0.001 x peak learning rate | 3.444 ± 0.002     |
> |      0.01 x peak learning rate  | 3.442 ± 0.002     |
> |      0.1 x peak learning rate   | 3.440 ± 0.002      |
>
> This experiment suggests that the precise level of final decay has little impact on the quality of the trained model, with no statistically significant difference between decay to 1% and decay to 10%. Therefore, we do not believe the final decay level affects the scaling laws we obtain.

---

> > ### Comment · Reviewer_comx · 2024-08-11
> >
> > Thank you for your response, and for the detailed discussion of the conjecture in Hoffmann et al. I maintain my positive assessment of the work.

---

> > > ### Author Response · Authors · 2024-08-11
> > >
> > > We are glad that our discussion of the conjecture in Hoffmann et al. was helpful and we are sure including it in the revision will also help future readers - thank you for bringing it up! The interpretation of Hoffmann et al.’s conjecture was the main weakness pointed out in the review. Now that it is resolved, and the two additional questions are answered, would you be willing to increase your score?

---

> ### Comment · Reviewer_comx · 2024-08-13
>
> I think that my current score is appropriate. Thank you.

---

### Official Review · Reviewer_szos · 2024-07-15

**Soundness:** 4
**Presentation:** 4
**Contribution:** 3
**Rating:** 8
**Confidence:** 4

**Summary:**

- Identifies and eliminates discrepancies between the scaling laws of Kaplan et al and Hoffman et al via three interventions: accounting for un-embed layer FLOPs, correcting warmup hyperparameters, and tuning optimizer hyperparameters.
- Also derives scaling laws for optimal learning rate and batch size as a function of model size

**Strengths:**

- The discrepancy between the scaling laws of Kaplan et al and Hoffman et al has been a major source of confusion/frustration in scaling law research. This paper conclusively identifies + mitigates this discrepancy through multiple targeted interventions.
- They also reproduce these results on two datasets, indicating that this discrepancy is not specific to the pretraining dataset.
- Section 4 provides complementary findings that are helpful for speeding up scaling law experiments: scaling laws for optimal learning rate and batch size, role of AdamW $\beta_2$, constant learning rate schedule suffices to reproduce Chinchilla scaling laws (albeit with higher loss)

**Weaknesses:**

The main limitation of this paper (as noted in the discussion) is that the range of models considered is fairly small compared to Hoffman et al. due to compute constraints. Nevertheless, it would be useful to have some sanity checks for checking if the estimated scaling laws for loss, learning rate, and batch size extrapolate reliably to 3B and 7B scale models.

**Questions:**

- Why did you opt for the IsoFlop estimation approach (approach 2 in chinchilla) rather than the arguably more direct "approach 3" in the chinchilla paper?
- In Appendix B, you mention that the main non-negligible source of error in approximating FLOPs with 6ND is the attention-related computation. This error can be quite non-trivial for small models with large context lengths, but would be small for large models. So, how would the scaling law exponents change if the FLOP estimator accounts for attention flops?

---

> ### Author Rebuttal · Authors · 2024-08-06
>
> We thank the reviewer for the useful questions and the encouraging comments, recognizing that our work conclusively identifies and mitigates the discrepancy between the two scaling laws, and appreciating the value of findings for speeding up scaling law experiments. Below, we address the questions brought up in the review.
>
> **Question 1: Approach 3 in Hoffmann et al.**
> In our opinion, Approach 3 is less direct because it arrives at the scaling law for $N^\star(C)$ via an assumption about how the loss depends on $N$ and $D$. In contrast, Approach 2 and our approach directly estimate $N^\star(C)$ and fit a scaling law to it, without requiring additional assumptions on the structure of the loss. Moreover, Besiroglu et al. [7] found that Approach 3 is sensitive to the curve-fitting method which has led to inaccurate conclusions in Hoffmann et al. Therefore we decided not to use Approach 3.
>
> **Question 2: Attention FLOPs**
> The scaling law exponents do not change significantly when we account for the attention FLOPS - see Figure 6 in the appendix and discussion in lines 531–551.
>
> **Extending experiments to larger models**
> Training a model with $N=3B$ with $\rho=16.5$ (our compute optimal tokens-to-parameters ratio) would require roughly 9e20 FLOPs, which is ~36 times larger than the largest individual training run in this project. A 7B model would be 200 times more expensive. Since verifying the optimality of hyperparameters requires several such training runs, such experiments are beyond our current resources. Consequently, we conducted a more modest scale-up to ~8e19 FLOPs, which is ~4 times the largest budget in our original FLOP grid. As discussed in our response to Reviewer Kjsg, our predicted learning rate and batch size are near-optimal even at that scale. Furthermore, as Figure 3 in the attached pdf shows, the loss remains fairly predictable as well.

---

> > ### Comment · Reviewer_szos · 2024-08-09
> >
> > Ok, thank you. I'd like to keep my score as is, great work!

---

### Author Rebuttal · Authors · 2024-08-07

We attach a file containing the relevant figures for the rebuttal.

---

### Decision · Program_Chairs · 2024-09-25

**Decision:**

Accept (spotlight)

**Comment:**

This paper examines the discrepancies between the two most influential scaling law papers in the literature (Kaplan et al. and Hoffmann et al.). The authors do a thorough job and manage to reconcile the conflicts in the prior work. All reviewers support acceptance and the AC agrees. This paper will be a good addition to the literature.